# ACCELERATING SEMIDEFINITE PROGRAMMING BEYOND LIMIT: ADMM WITH TUNE-FREE OPERATOR STEPSIZE

## ABSTRACT

In this work, we significantly alleviate the long-standing scalability issue of semidefinite programming (SDP), by equipping a novel tune-free operator stepsize to the alternating direction method of multipliers (ADMM) optimizer. To our best knowledge, this is the first operator stepsize in the context of SDP. More importantly, it is tune-free and computationally cheap (defined on dot product). Preliminary tests show that our operator ADMM surpasses the acceleration limit of the standard scalar version (limit found via grid search), i.e., our operator stepsize can outperform an arbitrarily fine tuned scalar one.

## 1 INTRODUCTION

Semidefinite programming (SDP) is widely recognized as one of the most important breakthroughs in the last century, with wide applications across fields, including machine learning, control, robotics, and communications. However, there exists a long-standing obstacle for SDP to gain further popularity — the scalability issue. It arises in middle or large scale problems, where an exponentially growing computation cost with data dimension is generally unacceptable. How to improve the scalability has been intensively studied, with several main directions: (i) exploiting structures, e.g., sparsity, symmetry, low-rankness etc.; (ii) approximation via linear and second-order cone programs. (iii) augmented Lagrangian methods, such as Newton-CG and alternating direction method of multipliers (ADMM); A comprehensive survey on scalability can be found in Majumdar et al. (2020). This manuscript will focus on the ADMM approach.

To start, we briefly review some history. SDP was developed as a generalization of linear programming (LP). The first polynomial-time solver of LP was introduced by Karmarkar (1984), termed the interior-point method (IPM). Later, Nesterov & Nemirovsky (1988); Nesterov & Nemirovskii (1994) extended it to any convex program, provided that the function is self-concordant. SDP can easily satisfy this condition, and hence been considered not much harder to solve than LP Vandenberghe & Boyd (1996).

Despite the great successes of IPMs, they are in general not well-suited for problems of middle or large scale. This is related to their second-order nature, where an inverse of the Hessian matrix is required, or at least an approximate inverse. Such an inverse operation is highly expensive for a large size variable. Even worse, the Hessian matrix is in general dense and rarely admits some structures like sparsity to reduce the computation cost. In the literature, employing first-order algorithms Beck (2017), Teboulle (2018) is considered one of the most promising directions. An outstanding candidate is ADMM Glowinski & Marroco (1975); Gabay & Mercier (1976). It has become increasingly popular, largely owes to a comprehensive survey by Boyd et al. (2011). In fact, one may already encounter ADMM, except under a different name. In recent years, many well-known algorithms have been revealed as equivalent to ADMM, such as the Douglas-Rachford Splitting (DRS) Lions & Mercier (1979); Douglas & Rachford (1956) and the Primal-Dual Hybrid Gradient (PDHG) method Pock et al. (2009); Esser et al. (2010); Chambolle & Pock (2011); O'Connor & Vandenberghe (2020).

The procedures for the first-order algorithms to solve SDP are largely similar, mainly differ on how to guarantee the solution being positive semidefinite (PSD). There are 3 typical strategies. (i) Directly define the variable in a quadratic form $\boldsymbol{R}^T \boldsymbol{R}$, which is always PSD, see e.g. Burer & Monteiro (2003; 2005); Wang et al. (2023). However, its efficiency highly depends on the dimension of $\boldsymbol{R}$, i.e., only efficient if it is low-rank. (ii) Enforcing PSD by a projected variable, denoted as $\Pi_{\mathbb{S}_+}(\boldsymbol{X})$. The success owes to that projector $\Pi_{\mathbb{S}_+}$ is strongly semi-smooth Sun & Sun (2002), and an inexact

semi-smooth Newton-CG method can apply, see e.g. Zhao et al. (2010). (iii) The last approach is via ADMM, which is a general framework that can apply to general convex problems, even some non-convex issues, see Boyd et al. (2011). Its application to SDP is well studied in Wen et al. (2010).

Our work corresponds to the ADMM method, most related to Wen et al. (2010). We achieved significant advances. (i) To our best knowledge, ours is the first operator stepsize in the context of SDP, not limited to the ADMM solver. For example, a diagonal matrix stepsize is not applicable here (no closed-form iterates). (ii) Our operator is specially designed, inspired by the Schur complement lemma. It enjoys the benefits of closed-form ADMM iterates and low computational cost (defined on dot product). (iii) Our operator stepsize is tune-free. It will be automatically updated based on a certain degree-4 polynomial. Numerically, we observed significant advantages compared to the empirical choice of scalar stepsize 1 and 1.6, as suggested in Wen et al. (2010). Even more, we performed a grid search to find the best scalar stepsize choice (least iteration number complexity sense), which can be viewed as the acceleration limit (not a priori knowledge). Preliminary tests show that our operator stepsize has surpassed such a limit.

For notations, $\|\cdot\|$ denotes the Euclidean norm, induced by the inner product $\langle\cdot,\cdot\rangle$. By $\circ$ we denote the operator composition. The uppercase bold, lowercase bold, and not bold letters are used for matrices, vectors, and scalars, respectively.

## 1.1 ADMM FRAMEWORK

To start, we introduce the general ADMM framework. It involves two sub-problems that typically admit closed-form solutions for a scalar stepsize, but often not when generalized to an operator one.

Consider a general convex program:

$$\begin{aligned}
\operatorname*{minimize}_{\boldsymbol{x},\boldsymbol{z}} \quad & f(\boldsymbol{x}) + g(\boldsymbol{z}), \\
\text{subject to} \quad & \mathcal{A}\boldsymbol{x} - \mathcal{B}\boldsymbol{z} = \boldsymbol{c},
\end{aligned} \tag{1.1}$$

with functions $f, g$ being convex, closed and proper (lower semi-continuous) and bounded linear operators $\mathcal{A}, \mathcal{B}$ being injective. A solution is assumed exists.

The standard ADMM iterates, with a scalar stepsize $\gamma > 0$, are

$$\begin{aligned}
\boldsymbol{x}^{k+1} &= \operatorname*{argmin}_{\boldsymbol{x}} \ f(\boldsymbol{x}) + \frac{\gamma}{2}\|\mathcal{A}\boldsymbol{x} - \mathcal{B}\boldsymbol{z}^k - \boldsymbol{c} + \boldsymbol{\lambda}^k/\gamma\|^2, \\
\boldsymbol{z}^{k+1} &= \operatorname*{argmin}_{\boldsymbol{z}} \ g(\boldsymbol{z}) + \frac{\gamma}{2}\|\mathcal{A}\boldsymbol{x}^{k+1} - \mathcal{B}\boldsymbol{z} - \boldsymbol{c} + \boldsymbol{\lambda}^k/\gamma\|^2, \\
\boldsymbol{\lambda}^{k+1} &= \boldsymbol{\lambda} + \gamma\big(\mathcal{A}\boldsymbol{x}^{k+1} - \mathcal{B}\boldsymbol{z}^{k+1} - \boldsymbol{c}\big), \qquad\qquad\text{(standard)}
\end{aligned}$$

The above can be generalized to an operator stepsize,

$$\begin{aligned}
\boldsymbol{x}^{k+1} &= \operatorname*{argmin}_{\boldsymbol{x}} \ f(\boldsymbol{x}) + \frac{1}{2}\|\mathcal{A}\boldsymbol{x} - \mathcal{B}\boldsymbol{z}^k - \boldsymbol{c} + \mathcal{M}^{-1}\boldsymbol{\lambda}^k\|^2_{\mathcal{M}}, \\
\boldsymbol{z}^{k+1} &= \operatorname*{argmin}_{\boldsymbol{z}} \ g(\boldsymbol{z}) + \frac{1}{2}\|\mathcal{A}\boldsymbol{x}^{k+1} - \mathcal{B}\boldsymbol{z} - \boldsymbol{c} + \mathcal{M}^{-1}\boldsymbol{\lambda}^k\|^2_{\mathcal{M}}, \\
\boldsymbol{\lambda}^{k+1} &= \boldsymbol{\lambda}^k + \mathcal{M}\big(\mathcal{A}\boldsymbol{x}^{k+1} - \mathcal{B}\boldsymbol{z}^{k+1} - \boldsymbol{c}\big), \qquad\qquad\text{(generalized)}
\end{aligned}$$

where $\mathcal{M} \succ 0$ is positive definite, and where $\|\boldsymbol{v}\|_{\mathcal{M}} = \sqrt{\langle\boldsymbol{v},\mathcal{M}\boldsymbol{v}\rangle}$ is known as the $\mathcal{M}$-norm.

Owing to $\mathcal{M}$ being positive definite, the decomposition $\mathcal{M} = \mathcal{S} \circ \mathcal{S}$ always exists. We will directly discuss the selection of $\mathcal{S}$, which is instantly transferable to $\mathcal{M}$.

## 1.2 SEMIDEFINITE PROGRAMMING

Semidefinite programming (SDP) include two standard forms, see Vandenberghe & Boyd (1996).

• (i) The standard primal SDP, which minimizes a linear function subject to a linear matrix inequality,

$$\begin{aligned}
\operatorname*{minimize}_{\boldsymbol{x}} \quad & \langle\boldsymbol{c},\boldsymbol{x}\rangle, \\
\text{subject to} \quad & \boldsymbol{A}_0 + \sum_{i=1}^{m} x_i\boldsymbol{A}_i \succeq 0. \qquad\qquad\text{(primal)}
\end{aligned}$$

with $\boldsymbol{A}_i \in \mathbb{S}^n$, $i = 0, 1, \ldots, m$ being symmetric matrices.

- (ii) The standard dual SDP, which is written in a matrix variable,

$$\underset{\boldsymbol{X}}{\text{minimize}} \quad \langle \boldsymbol{A}_0, \boldsymbol{X} \rangle,$$

$$\text{subject to} \quad \langle \boldsymbol{A}_i, \boldsymbol{X} \rangle = c_i, \quad i = 1, \ldots, m,$$

$$\boldsymbol{X} \succeq 0. \tag{dual}$$

The ADMM steps for solving the above two formulations will be similar, detailed below.

## 1.3 ADMM SOLVER

Here, we apply the abstract ADMM framework to solve the two SDP problems, equation primal and equation dual. We can compactly write them into

$$\underset{\boldsymbol{X}, \boldsymbol{Z}}{\text{minimize}} \quad f(\boldsymbol{X}) + \delta_{\mathbb{S}^n_+}(\boldsymbol{Z}),$$

$$\text{subject to} \quad \mathcal{A}\boldsymbol{X} = \boldsymbol{Z}, \tag{1.2}$$

with $\boldsymbol{X} \in \mathbb{R}^{n \times n}$, $\boldsymbol{Z} \in \mathbb{R}^{n \times n}$ being matrix variables (can reduce to vectors), where $\delta_{\mathbb{S}^n_+}$ denotes the indicator function on the semidefinite cone $\mathbb{S}^n_+$. The above unified framework can be specified into

(i) equation primal via

$$f(\boldsymbol{x}) = \langle \boldsymbol{c}, \boldsymbol{x} \rangle, \quad \mathcal{A}\boldsymbol{x} = \boldsymbol{A}_0 + \sum_{i=1}^{m} x_i \boldsymbol{A}_i, \tag{1.3}$$

with $\boldsymbol{x} \in \mathbb{R}^n$, $\text{dom } f = \mathbb{R}^n$.

(ii) equation dual via

$$f(\boldsymbol{X}) = \langle \boldsymbol{A}_0, \boldsymbol{X} \rangle, \quad \text{dom } f = \{ \boldsymbol{X} \in \mathbb{S}^n \,|\, \langle \boldsymbol{A}_i, \boldsymbol{X} \rangle = c_i, \, \forall i \}, \tag{1.4}$$

and $\mathcal{A} = \mathcal{I}$ vanishes (identity operator).

### 1.3.1 IMPLEMENTATION DETAILS (SCALAR CASE)

Here, we present the ADMM closed-form iterates for solving equation 1.2, given a scalar stepsize.

(i) For equation primal, its X-update is given by

$$\boldsymbol{x}^{k+1} = \left( \bar{\boldsymbol{A}}^T \bar{\boldsymbol{A}} \right)^{-1} \left( \bar{\boldsymbol{A}}^T \left( \boldsymbol{Z}^k - \boldsymbol{\Lambda}^k / \gamma - \bar{\boldsymbol{A}}_0 \right) - \boldsymbol{c} / \gamma \right), \tag{1.5}$$

where $\bar{\boldsymbol{A}} = [\text{vec}(\boldsymbol{A}_1), \cdots, \text{vec}(\boldsymbol{A}_m)] \in \mathbb{R}^{n^2 \times m}$, $\bar{\boldsymbol{A}}_0 = \text{vec}(\boldsymbol{A}_0) \in \mathbb{R}^{n^2 \times 1}$.

(ii) For equation dual, its X-update is given by solving the following KKT system:

$$\begin{bmatrix} \gamma \boldsymbol{I} & \bar{\boldsymbol{A}} \\ \bar{\boldsymbol{A}}^T & \boldsymbol{0} \end{bmatrix} \begin{bmatrix} \text{vec}(\boldsymbol{X}^{k+1}) \\ \boldsymbol{\mu} \end{bmatrix} = \begin{bmatrix} \text{vec}(\gamma \boldsymbol{Z}^k - \boldsymbol{\Lambda}^k - \boldsymbol{A}_0) \\ \boldsymbol{c} \end{bmatrix}, \tag{1.6}$$

which is an overdetermined system that is instantly solvable via a pseudo inverse.

The other iterates are the same for the primal and dual SDP. The Z-update is

$$\boldsymbol{Z}^{k+1} = \Pi_{\mathbb{S}^n_+} \left( \mathcal{A}\boldsymbol{X}^{k+1} + \boldsymbol{\Lambda}^k / \gamma \right), \tag{1.7}$$

which is a projection, setting all the negative eigenvalues to zeros. At last,

$$\boldsymbol{\Lambda}^{k+1} = \boldsymbol{\Lambda}^k + \gamma \left( \mathcal{A}\boldsymbol{X}^{k+1} - \boldsymbol{Z}^{k+1} \right). \tag{1.8}$$

### 1.3.2 GENERALIZATION CHALLENGE

The main challenge for employing an operator stepsize is that — in general, the Z-update no longer admits a closed-form iterate, i.e., the following:

$$\boldsymbol{Z}^{k+1} = \underset{\boldsymbol{z}}{\text{argmin}} \; \delta_{\mathbb{S}^n_+}(\boldsymbol{Z}) + \frac{1}{2} \| \mathcal{A}\boldsymbol{X}^{k+1} - \boldsymbol{Z} + \mathcal{M}^{-1}\boldsymbol{\Lambda}^k \|_{\mathcal{M}}^2,$$

$$= \underset{\boldsymbol{z}}{\text{argmin}} \; \delta_{\mathbb{S}^n_+}(\boldsymbol{Z}) + \frac{1}{2} \| \mathcal{S}\mathcal{A}\boldsymbol{X}^{k+1} - \mathcal{S}\boldsymbol{Z} + \mathcal{S}^{-1}\boldsymbol{\Lambda}^k \|^2. \tag{1.9}$$

does not admit a closed-form solution in general.

## 1.4 OUR CONTRIBUTION

Our contribution involves two main aspects. (i) First, we propose the following specially designed operator stepsize, inspired by the Schur complement lemma, see Proposition 2.1:

$$\mathcal{S} = \left[ \begin{array}{cc} \sqrt{\frac{\gamma_1}{\gamma_2}}\, \mathbf{1}_1 & \sqrt{\gamma_1}\, \mathbf{1}_0 \\ \sqrt{\gamma_1}\, \mathbf{1}_0 & \sqrt{\gamma_1 \gamma_2}\, \mathbf{1}_2 \end{array} \right], \quad \mathcal{S}^{-1} = \left[ \begin{array}{cc} \sqrt{\frac{\gamma_2}{\gamma_1}}\, \mathbf{1}_1 & \frac{1}{\sqrt{\gamma_1}}\, \mathbf{1}_0 \\ \frac{1}{\sqrt{\gamma_1}}\, \mathbf{1}_0 & \frac{1}{\sqrt{\gamma_1 \gamma_2}}\, \mathbf{1}_2 \end{array} \right]. \tag{1.10}$$

where $\mathbf{1}_1 \in \mathbb{S}^m$, $\mathbf{1}_0 \in \mathbb{R}^{m \times (n-m)}$, $\mathbf{1}_2 \in \mathbb{S}^{n-m}$ are ones matrices.

• It is computationally cheap, due to defined on element-wise multiplication (a.k.a. dot product). Particularly, its inverse $\mathcal{S}^{-1}$ does not require computation, owing to the above explicit form.

• It addresses the closed-form iterates challenge, aforementioned in Section 1.3.2. Specifically, equation 1.9 admits the following closed-form solution:

$$\boldsymbol{Z}^{k+1} = \mathcal{S}^{-1} \Pi_{\mathbb{S}^n_+} \left( \mathcal{S} \mathcal{A} \boldsymbol{X}^{k+1} + \mathcal{S}^{-1} \boldsymbol{\Lambda}^k \right). \tag{1.11}$$

(ii) The above operator stepsize $\mathcal{S}$ does not need any tuning. It will be automatically calculated via the closed-form root of a degree-4 polynomial. Moreover, such a stepsize update can be early stopped to save some runtime.

Below, we summarize our operator ADMM algorithm, with slightly different steps for equation primal and equation dual. They may be simplified if some tailored structures exploited.

---

**Algorithm 1** SDP via operator ADMM (standard primal version)

---

**Input:** Set $\boldsymbol{Z}^0 = \mathbf{0}$, $\boldsymbol{\Lambda}^0 = \mathbf{0}$, $\mathcal{S}_0 = \mathbf{1}$.
1: **while** iterates not converged **do**
2:

$$\boldsymbol{x}^{k+1} \leftarrow \left( \tilde{\boldsymbol{A}}^T \tilde{\boldsymbol{A}} \right)^{-1} \left( \tilde{\boldsymbol{A}}^T \left( \mathcal{S}_k \boldsymbol{Z}^k - \mathcal{S}_k^{-1} \boldsymbol{\Lambda}^k - \tilde{\boldsymbol{A}}_0 \right) - \boldsymbol{c} \right),$$

$$\boldsymbol{Z}^{k+1} \leftarrow \mathcal{S}_k^{-1} \Pi_{\mathbb{S}^n_+} \left( \tilde{\boldsymbol{A}}_0 + \mathrm{mat}(\tilde{\boldsymbol{A}} \boldsymbol{x}^{k+1}) + \mathcal{S}_k^{-1} \boldsymbol{\Lambda}^k \right),$$

$$\boldsymbol{\Lambda}^{k+1} \leftarrow \boldsymbol{\Lambda}^k + \mathcal{S}_k \left( \tilde{\boldsymbol{A}}_0 + \mathrm{mat}(\tilde{\boldsymbol{A}} \boldsymbol{x}^{k+1}) - \mathcal{S}_k \boldsymbol{Z}^{k+1} \right), \tag{primal}$$

where $\tilde{\boldsymbol{A}}_0 = \mathrm{vec}(\mathcal{S}_k \boldsymbol{A}_0)$, $\tilde{\boldsymbol{A}} = [\mathrm{vec}(\mathcal{S}_k \boldsymbol{A}_1), \cdots, \mathrm{vec}(\mathcal{S}_k \boldsymbol{A}_m)]$.
3:    **operator adaption:** Compute $\mathcal{S}_{k+1}$ via Corollary 3.2.
4: **end while**
**Output:** primal solution $\boldsymbol{x}^\star$, dual solution $\boldsymbol{\Lambda}^\star$.

---

**Algorithm 2** SDP via operator ADMM (standard dual version)

---

**Input:** Set $\boldsymbol{Z}^0 = \mathbf{0}$, $\boldsymbol{\Lambda}^0 = \mathbf{0}$, $\mathcal{S}_0 = \mathbf{1}$.
1: **while** iterates not converged **do**
2:    X-update via the following KKT system (pseudo inverse):

$$\left[ \begin{array}{cc} \mathcal{S}_k \circ \mathcal{S}_k(\boldsymbol{I}) & \bar{\boldsymbol{A}} \\ \bar{\boldsymbol{A}}^T & \mathbf{0} \end{array} \right] \left[ \begin{array}{c} \mathrm{vec}(\boldsymbol{X}^{k+1}) \\ \boldsymbol{\mu} \end{array} \right] = \left[ \begin{array}{c} \mathrm{vec}\left( \mathcal{S}_k \circ \mathcal{S}_k(\boldsymbol{Z}^k) - \boldsymbol{\Lambda}^k - \boldsymbol{A}_0 \right) \\ \boldsymbol{c} \end{array} \right], \tag{dual}$$

where $\bar{\boldsymbol{A}} = [\mathrm{vec}(\boldsymbol{A}_1), \cdots, \mathrm{vec}(\boldsymbol{A}_m)]$. The rest iterates are

$$\boldsymbol{Z}^{k+1} \leftarrow \mathcal{S}_k^{-1} \Pi_{\mathbb{S}^n_+} \left( \mathcal{S}_k \boldsymbol{X}^{k+1} + \mathcal{S}_k^{-1} \boldsymbol{\Lambda}^k \right),$$

$$\boldsymbol{\Lambda}^{k+1} \leftarrow \boldsymbol{\Lambda}^k + \mathcal{S}_k \circ \mathcal{S}_k \left( \boldsymbol{X}^{k+1} - \boldsymbol{Z}^{k+1} \right). \tag{cont.}$$

3:    **operator adaption:** Compute $\mathcal{S}_{k+1}$ via Corollary 3.2.
4: **end while**
**Output:** primal solution $\boldsymbol{X}^\star$, dual solution $\boldsymbol{\Lambda}^\star$.

---

## 2 CLOSED-FORM GUARANTEE (OPERATOR CASE)

Here, we address the closed-form issue of an operator stepsize, aforementioned in Sec. 1.3.2.

### 2.1 OPERATOR DESIGN

It begins with the following insight:

**Lemma 2.1** (guideline). *Given any invertible operator $\mathcal{S}$, with its inverse denoted as $\mathcal{S}^{-1}$. Suppose the following holds:*

$$\mathcal{S}^{-1}(\boldsymbol{Z}) \in \mathbb{S}_+^n, \qquad \forall \boldsymbol{Z} \in \mathbb{S}_+^n. \tag{2.1}$$

*Then, a closed-form solution is available:*

$$\mathcal{S}^{-1} \circ \Pi_{\mathbb{S}_+^n}(\mathcal{S}\boldsymbol{V}) = \underset{\boldsymbol{Z}}{argmin}\ \delta_{\mathbb{S}_+^n}(\boldsymbol{Z}) + \frac{1}{2}\|\boldsymbol{Z} - \boldsymbol{V}\|_{\mathcal{M}}^2,$$

$$= \underset{\boldsymbol{Z}}{argmin}\ \delta_{\mathbb{S}_+^n}(\boldsymbol{Z}) + \frac{1}{2}\|\mathcal{S}\boldsymbol{Z} - \mathcal{S}\boldsymbol{V}\|^2, \tag{2.2}$$

*where $\mathcal{M} = \mathcal{S} \circ \mathcal{S}$.*

Following from above, all we need is to design an operator satisfying equation 2.1.

**Proposition 2.1** (operator design). *Given scalars $\gamma_1, \gamma_2 > 0$ and any integer $m \in \{1, 2, \ldots, n-1\}$. Let operator $\mathcal{S}$ be defined as*

$$\mathcal{S}(\boldsymbol{V}) = \begin{bmatrix} \sqrt{\frac{\gamma_1}{\gamma_2}}\,\mathbf{1}_1 & \sqrt{\gamma_1}\,\mathbf{1}_0 \\ \sqrt{\gamma_1}\,\mathbf{1}_0 & \sqrt{\gamma_1\gamma_2}\,\mathbf{1}_2 \end{bmatrix} \odot \boldsymbol{V}, \tag{2.3}$$

*and its inverse being*

$$\mathcal{S}^{-1}(\boldsymbol{V}) = \begin{bmatrix} \sqrt{\frac{\gamma_2}{\gamma_1}}\,\mathbf{1}_1 & \frac{1}{\sqrt{\gamma_1}}\,\mathbf{1}_0 \\ \frac{1}{\sqrt{\gamma_1}}\,\mathbf{1}_0 & \frac{1}{\sqrt{\gamma_1\gamma_2}}\,\mathbf{1}_2 \end{bmatrix} \odot \boldsymbol{V}, \tag{2.4}$$

*where $\mathbf{1}_1 \in \mathbb{S}^m$, $\mathbf{1}_0 \in \mathbb{R}^{m \times (n-m)}$, $\mathbf{1}_2 \in \mathbb{S}^{n-m}$, and where $\mathbf{1}$ denotes the ones matrix (i.e., all entries being 1), by $\odot$ the element-wise multiplication.*

*Then,*

$$\mathcal{S}^{-1} \circ \Pi_{\mathbb{S}_+^n}(\mathcal{S}\boldsymbol{V}) = \underset{\boldsymbol{Z}}{argmin}\ \delta_{\mathbb{S}_+^n}(\boldsymbol{Z}) + \frac{1}{2}\|\mathcal{S}\boldsymbol{Z} - \mathcal{S}\boldsymbol{V}\|^2. \tag{2.5}$$

*Remarks* 2.1 (partitioning choice). Above, any integer $m \in \{1, 2, \ldots, n-1\}$ is feasible. However, the algorithm performance does change with $m$ (but not too sensitive). Empirically, we find the choice $m = n - 1$ typically works well, and we set it as the default.

### 2.1.1 ADMM IMPLEMENTATION (OPERATOR CASE)

Equipping the above operator stepsize $\mathcal{S}$ to ADMM, we arrive at the following iterates (for solving equation 1.2):

$$\boldsymbol{X}^{k+1} = \underset{\boldsymbol{X}}{argmin}\ f(\boldsymbol{X}) + \frac{1}{2}\|\mathcal{S}\mathcal{A}\boldsymbol{X} - \mathcal{S}\boldsymbol{Z}^k + \mathcal{S}^{-1}\boldsymbol{\Lambda}^k\|^2,$$

$$\boldsymbol{Z}^{k+1} = \mathcal{S}^{-1}\Pi_{\mathbb{S}_+^n}\left(\mathcal{S}\mathcal{A}\boldsymbol{X}^{k+1} + \mathcal{S}^{-1}\boldsymbol{\Lambda}^k\right),$$

$$\boldsymbol{\Lambda}^{k+1} = \boldsymbol{\Lambda}^k + \mathcal{S}\left(\mathcal{S}\mathcal{A}\boldsymbol{X}^{k+1} - \mathcal{S}\boldsymbol{Z}^{k+1}\right). \tag{2.6}$$

The above X-update can be further written into a closed form. Specifically,

(i) for equation primal, we have

$$\boldsymbol{x}^{k+1} = \left(\tilde{\boldsymbol{A}}^T\tilde{\boldsymbol{A}}\right)^{-1}\left(\tilde{\boldsymbol{A}}^T\left(\mathcal{S}\boldsymbol{Z}^k - \mathcal{S}^{-1}\boldsymbol{\Lambda}^k - \tilde{\boldsymbol{A}}_0\right) - \boldsymbol{c}\right), \tag{2.7}$$

where $\tilde{\boldsymbol{A}} = [\text{vec}(\mathcal{S}\boldsymbol{A}_1), \cdots, \text{vec}(\mathcal{S}\boldsymbol{A}_m)] \in \mathbb{R}^{n^2 \times m}$, and where $\tilde{\boldsymbol{A}}_0 = \text{vec}(\mathcal{S}\boldsymbol{A}_0)$.

• For equation dual, the X-update closed-form is given by solving the following KKT system:

$$\begin{bmatrix} \mathcal{S} \circ \mathcal{S}(\boldsymbol{I}) & \bar{\boldsymbol{A}} \\ \bar{\boldsymbol{A}}^T & \boldsymbol{0} \end{bmatrix} \begin{bmatrix} \text{vec}(\boldsymbol{X}^{k+1}) \\ \boldsymbol{\mu} \end{bmatrix} = \begin{bmatrix} \text{vec}\left(\mathcal{S} \circ \mathcal{S}(\boldsymbol{Z}^k) - \boldsymbol{\Lambda}^k - \boldsymbol{A}_0\right) \\ \boldsymbol{c} \end{bmatrix}, \tag{2.8}$$

where $\bar{\boldsymbol{A}} = [\text{vec}(\boldsymbol{A}_1), \cdots, \text{vec}(\boldsymbol{A}_m)] \in \mathbb{R}^{n^2 \times m}$. It is an overdetermined system that is instantly solvable via a pseudo inverse. ($\boldsymbol{\mu}$ is an auxiliary variable that will be omitted.)

## 3 OPERATOR STEPSIZE SELECTION

Here, we show how to select our operator stepsize automatically. It involves two steps. First, minimize an upper bound, which yields a theoretical optimal choice (not a priori knowledge). Then, we approximate such a choice successively.

### 3.1 THEORETICAL CHOICE

To start, we need a characterization of the ADMM convergence rate. It is first established in He & Yuan (2015) through variational inequality. Below, we will adopt a recent fixed-point argument from Ryu & Yin (2022), which is slightly more convenient.

**Lemma 3.1.** *(Ryu & Yin, 2022, Theorem 1) ADMM admits the following worst-case convergence rate:*

$$\|\boldsymbol{\zeta}^{k+1} - \boldsymbol{\zeta}^k\|^2 \leq \frac{1}{k+1}\|\boldsymbol{\zeta}^\star - \boldsymbol{\zeta}^0\|^2, \tag{3.1}$$

*where initialization $\boldsymbol{\zeta}^0$ can be arbitrary.*

Our ADMM iterates as in equation 2.6 corresponds to the above fixed-point view, via

$$\boldsymbol{\zeta}^{k+1} = \mathcal{S}\mathcal{A}\boldsymbol{X}^{k+1} + \mathcal{S}^{-1}\boldsymbol{\Lambda}^k. \tag{3.2}$$

**Corollary 3.1.** *Under zero initialization $\boldsymbol{X}^0 = \boldsymbol{Z}^0 = \boldsymbol{\Lambda}^0 = \boldsymbol{0}$, the worst-case optimal choice of our operator stepsize $\mathcal{S}$ can be determined via*

$$\underset{\mathcal{S}}{minimize} \quad \|\mathcal{S}\mathcal{A}\boldsymbol{X}^\star + \mathcal{S}^{-1}\boldsymbol{\Lambda}^\star\|^2, \tag{3.3}$$

### 3.1.1 SOLUTION DETAILS

Now, we solve the above problem. For the sake of light notation, denote $\hat{\boldsymbol{X}} = \mathcal{A}\boldsymbol{X}$. Also, define partitioning

$$\hat{\boldsymbol{X}} = \begin{bmatrix} \hat{\boldsymbol{X}}_1 & \hat{\boldsymbol{X}}_0 \\ \hat{\boldsymbol{X}}_0^\mathrm{T} & \hat{\boldsymbol{X}}_2 \end{bmatrix}, \quad \boldsymbol{\Lambda} = \begin{bmatrix} \boldsymbol{\Lambda}_1 & \boldsymbol{\Lambda}_0 \\ \boldsymbol{\Lambda}_0^\mathrm{T} & \boldsymbol{\Lambda}_2 \end{bmatrix}, \tag{3.4}$$

where $\hat{\boldsymbol{X}}_1, \boldsymbol{\Lambda}_1 \in \mathbb{S}^m$, $\hat{\boldsymbol{X}}_2, \boldsymbol{\Lambda}_2 \in \mathbb{S}^{n-m}$ are symmetric matrices, and where $\hat{\boldsymbol{X}}_0, \boldsymbol{\Lambda}_0 \in \mathbb{R}^{m \times (n-m)}$.

**Lemma 3.2.** *Invoke the definition of $\mathcal{S}$ in equation 2.3. equation 3.3 can be rewritten into*

$$\underset{\gamma_1, \gamma_2 > 0}{minimize} \frac{\gamma_1}{\gamma_2}\|\hat{\boldsymbol{X}}_1^\star\|^2 + \frac{\gamma_2}{\gamma_1}\|\boldsymbol{\Lambda}_1^\star\|^2 + \gamma_1\gamma_2\|\hat{\boldsymbol{X}}_2^\star\|^2 + \frac{1}{\gamma_1\gamma_2}\|\boldsymbol{\Lambda}_2^\star\|^2 + 2\gamma_1\|\hat{\boldsymbol{X}}_0^\star\|^2 + \frac{2}{\gamma_1}\|\boldsymbol{\Lambda}_0^\star\|^2. \tag{3.5}$$

The above admits closed-form solutions, related to the root of the polynomial below.

**Lemma 3.3** (polynomial root). *Consider the following degree-4 polynomial:*

$$a\rho^4 + b\rho^3 + d\rho + e = 0. \tag{3.6}$$

*Suppose all coefficients are real and $b, d$ not simultaneously equal 0. Then, it admits 4 closed-form roots as*

$$\rho = \begin{cases} \frac{1}{2}(-\frac{b}{2a} - u_4 - \sqrt{u_5 - u_6}), \\ \frac{1}{2}(-\frac{b}{2a} - u_4 + \sqrt{u_5 - u_6}), \\ \frac{1}{2}(-\frac{b}{2a} + u_4 - \sqrt{u_5 + u_6}), \\ \frac{1}{2}(-\frac{b}{2a} + u_4 + \sqrt{u_5 + u_6}), \end{cases} \tag{3.7}$$

*where*

$$u_4 = \sqrt{\frac{b^2}{4a^2} + u_3}, \quad u_5 = \frac{b^2}{2a^2} - u_3, \quad u_6 = -\frac{\frac{b^3}{a^3} + \frac{8d}{a}}{4u_4}, \tag{3.8}$$

*and where*

$$u_1 = \frac{\sqrt{27}}{2}(ad^2 + b^2 e), \ u_2 = u_1 + \sqrt{(bd - 4ae)^3 + u_1^2}, \ u_3 = \frac{1}{\sqrt{3}a}(\sqrt[3]{u_2} - \frac{bd - 4ae}{\sqrt[3]{u_2}}). \tag{3.9}$$

**Theorem 3.1.** *The worst-case optimal choice of stepsize $\mathcal{S}$, or equivalently the parameter pair $(\gamma_1, \gamma_2)$, is given by*

$$\gamma_1^\star = \sqrt{\frac{\gamma_2^\star \|\mathbf{\Lambda}_1^\star\|^2 + \frac{1}{\gamma_2^\star}\|\mathbf{\Lambda}_2^\star\|^2 + 2\|\mathbf{\Lambda}_0^\star\|^2}{\frac{1}{\gamma_2^\star}\|\hat{\mathbf{X}}_1^\star\|^2 + \gamma_2^\star\|\hat{\mathbf{X}}_2^\star\|^2 + 2\|\hat{\mathbf{X}}_0^\star\|^2}}. \tag{3.10}$$

*with $\gamma_2^\star$ being a positive root of the following degree-4 polynomial:*

$$\gamma_2^{\star 4}\|\hat{\mathbf{X}}_2^\star\|^2\|\mathbf{\Lambda}_1^\star\|^2 + \gamma_2^{\star 3}(\|\hat{\mathbf{X}}_2^\star\|^2\|\mathbf{\Lambda}_0^\star\|^2 + \|\hat{\mathbf{X}}_0^\star\|^2\|\mathbf{\Lambda}_1^\star\|^2) - \gamma_2^\star(\|\mathbf{\Lambda}_2^\star\|^2\|\hat{\mathbf{X}}_0^\star\|^2 + \|\mathbf{\Lambda}_0^\star\|^2\|\hat{\mathbf{X}}_1^\star\|^2)$$

$$- \|\mathbf{\Lambda}_2^\star\|^2\|\hat{\mathbf{X}}_1^\star\|^2 = 0, \tag{3.11}$$

*with a closed-form solution available via Lemma 3.3.*

### 3.2 PRACTICAL USE

The above involves certain optimal point information, hence not instantly useful in practice. To address it, we replace the optimal solutions (unknown) by the current iterates (known).

Similar approximation idea already appears in the machine learning field, but on a different issue, the importance sampling, see e.g. Yuan et al. (2016), (Rizk et al., 2022, Sec. IV. C).

**Corollary 3.2.** *The $(k + 1)$-th operator stepsize $\mathcal{S}_{k+1}$ can be determined via*

$$\gamma_1^{k+1} = \sqrt{\frac{\gamma_2^{k+1}\|\mathbf{\Lambda}_1^{k+1}\|^2 + \frac{1}{\gamma_2^{k+1}}\|\mathbf{\Lambda}_2^{k+1}\|^2 + 2\|\mathbf{\Lambda}_0^{k+1}\|^2}{\frac{1}{\gamma_2^{k+1}}\|\hat{\mathbf{X}}_1^{k+1}\|^2 + \gamma_2^{k+1}\|\hat{\mathbf{X}}_2^{k+1}\|^2 + 2\|\hat{\mathbf{X}}_0^{k+1}\|^2}}. \tag{3.12}$$

*with $\gamma_2^{k+1}$ being a positive root of the following degree-4 polynomial:*

$$\gamma_2^{k+1^4}\|\hat{\mathbf{X}}_2^{k+1}\|^2\|\mathbf{\Lambda}_1^{k+1}\|^2 + \gamma_2^{k+1^3}(\|\hat{\mathbf{X}}_2^{k+1}\|^2\|\mathbf{\Lambda}_0^{k+1}\|^2 + \|\hat{\mathbf{X}}_0^{k+1}\|^2\|\mathbf{\Lambda}_1^{k+1}\|^2)$$

$$- \gamma_2^{k+1}(\|\mathbf{\Lambda}_2^{k+1}\|^2\|\hat{\mathbf{X}}_0^{k+1}\|^2 + \|\mathbf{\Lambda}_0^{k+1}\|^2\|\hat{\mathbf{X}}_1^{k+1}\|^2) - \|\mathbf{\Lambda}_2^{k+1}\|^2\|\hat{\mathbf{X}}_1^{k+1}\|^2 = 0, \tag{3.13}$$

*with a closed-form solution available via Lemma 3.3.*

## 4 NUMERICAL EXAMPLES

In this section, we present two examples, arising from digital communication and machine learning.

### 4.1 BOOLEAN QUADRATIC PROGRAM

We start with the Boolean quadratic program. It is a fundamental problem in digital communication, particularly popular in circuit design.

Ideally, one would like to solve the following Boolean program:

$$\underset{\mathbf{x}}{\text{minimize}} \quad \|\mathbf{Ax} - \mathbf{b}\|^2,$$

$$\text{subject to} \quad x_i \in \{-1, 1\}, \quad i = 1, \dots, n, \tag{NP-hard}$$

with $\mathbf{x} \in \mathbb{R}^{n \times 1}, \mathbf{b} \in \mathbb{R}^{m \times 1}, \mathbf{A} \in \mathbb{R}^{m \times n}$. This problem is well-known to be NP-hard. In the literature, it is common to instead solving a semidefinite relaxed version, which we compactly written as

$$\underset{\mathbf{X}}{\text{minimize}} \quad \langle \mathbf{A}_0, \mathbf{X} \rangle,$$

$$\text{subject to} \quad \text{diag}(\mathbf{X}) = \mathbf{1},$$

$$\mathbf{X} \succeq 0, \tag{relaxed}$$

where $\boldsymbol{A}_0 = \begin{bmatrix} \boldsymbol{A}^{\mathrm{T}}\boldsymbol{A} & \boldsymbol{b} \\ \boldsymbol{b}^{\mathrm{T}} & 0 \end{bmatrix}$. This formulation corresponds to the standard dual form as in equation dual, and is solvable via our Algorithm 2.

However, we emphasize that our general solver cannot exploit any tailored structure. Here, the diagonal constraint is highly structured, and it would be a better idea to employ a tailored version, which is both simpler and more efficient, summarized below.

---

**Algorithm 3** relaxed BQP via operator ADMM (simplified version; tailored structure)

---

**Input:** Set $\boldsymbol{Z}^0 = \boldsymbol{0}$, $\boldsymbol{\Lambda}^0 = \boldsymbol{0}$, $\mathcal{S}_0 = \boldsymbol{1}$.
1: **while** iterates not converged **do**
2:

$$\boldsymbol{X}^{k+1} \leftarrow \boldsymbol{Z}^k - \mathcal{S}_k^{-1} \circ \mathcal{S}_k^{-1}\left(\boldsymbol{\Lambda}^k + \boldsymbol{A}_0\right),$$
$$\mathrm{diag}(\boldsymbol{X}^{k+1}) \leftarrow \boldsymbol{1},$$
$$\boldsymbol{Z}^{k+1} \leftarrow \mathcal{S}_k^{-1}\Pi_{\mathbb{S}_+^n}\left(\mathcal{S}_k\boldsymbol{X}^{k+1} + \mathcal{S}_k^{-1}\boldsymbol{\Lambda}^k\right),$$
$$\boldsymbol{\Lambda}^{k+1} \leftarrow \boldsymbol{\Lambda}^k + \mathcal{S}_k \circ \mathcal{S}_k\left(\boldsymbol{X}^{k+1} - \boldsymbol{Z}^{k+1}\right). \tag{4.1}$$

3:     **operator adaption:** Compute $\mathcal{S}_{k+1}$ via Corollary 3.2.
4: **end while**
**Output:** primal solution $\boldsymbol{X}^\star$, dual solution $\boldsymbol{\Lambda}^\star$.

---

#### 4.1.1 BEYOND THE LIMIT

Here, we compare our operator stepsize with the underlying best scalar choice. Such a best scalar is not a priori knowledge, and we find it by grid searching (under a fixed random number generator).

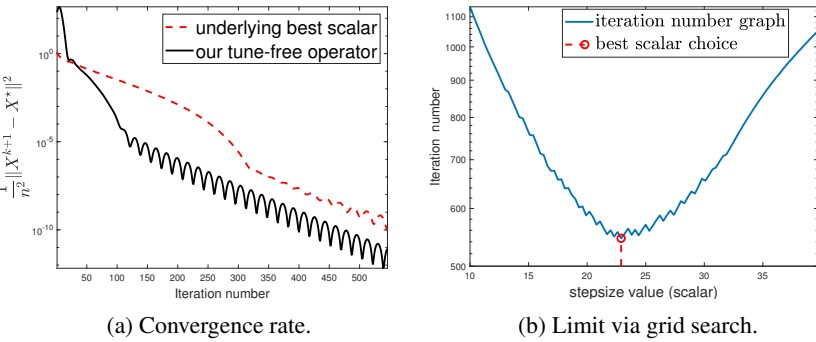

(a) Convergence rate.          (b) Limit via grid search.

Figure 1: Beyond the scalar-case limit, $m = n = 50$.

*Remarks* 4.1. We observed that our operator stepsize outperforms the best scalar choice by a noticeable margin. Particularly, ours is around $4\times$ faster to reach a moderate accuracy of $10^{-5}$.

*Remarks* 4.2. Additionally, we find that the best scalar varies rapidly, being highly sensitive to different data sizes and types. It appears impossible to make a direct guess.

#### 4.1.2 SCALABILITY

Here, we concern the scalability issue. We compare our operator with two empirical scalar stepsize choices, value 1 and 1.6, suggested in Wen et al. (2010). The algorithm will stop if a mean squared error threshold of $10^{-4}$ reached. The error is measured by comparing to the ground-truth, generated via CVX Grant & Boyd (2014).

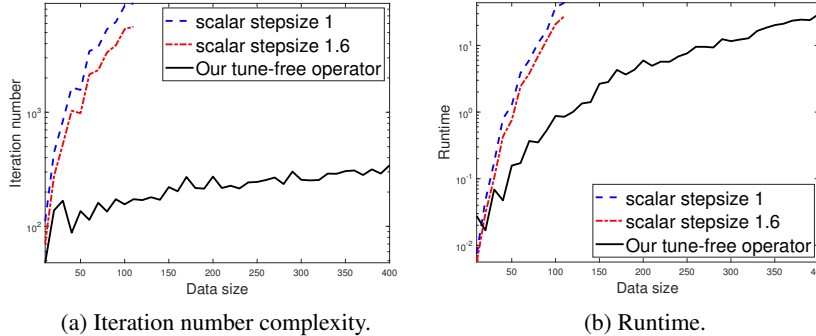

(a) Iteration number complexity.     (b) Runtime.

Figure 2: Scalability: our operator stepsize vs. fixed scalar 1 and 1.6.

*Remarks* 4.3. Figure 2a measures the iteration number complexity, and ours shows an overwhelming advantage, roughly $50\times$ acceleration for an $\mathbb{R}^{100\times 1}$ size variable, and much more when the data size further increases. Based on the curvature, ours does show a significantly better scalability.

*Remarks* 4.4. Figure 2b measures the CPU runtime by a 'tic toc' command in MATLAB. Ours has a marginal disadvantage at the start, but soon gains advantage and arrives at roughly $10\times$ acceleration for an $\mathbb{R}^{100\times 1}$ size variable. Our advantage increases consistently with data dimension, based on the curvature of the plot.

### 4.2 DISTANCE METRIC LEARNING

Here, we consider the distance metric learning problem in machine learning. A metric, by definition, needs to be positive semidefinite, hence well-fitted into our scope.

Below, we adopt the notation and data setup from Xing et al. (2002a). Consider finding a distance metric $\boldsymbol{A}$ via

$$\underset{\boldsymbol{A},\boldsymbol{Z}\in\mathbb{S}^m}{\text{minimize}} \sum_{(\boldsymbol{x}_i,\boldsymbol{x}_j)\in S} \|\boldsymbol{x}_i - \boldsymbol{x}_j\|_{\boldsymbol{A}}^2 - \log\left(\sum_{(\boldsymbol{x}_i,\boldsymbol{x}_j)\notin S} \|\boldsymbol{x}_i - \boldsymbol{x}_j\|_{\boldsymbol{A}}^2\right) + \delta_{\mathbb{S}_+^m}(\boldsymbol{Z}),$$

$$\text{subject to} \quad \boldsymbol{A} = \boldsymbol{Z}, \tag{4.2}$$

where $\boldsymbol{x}_i, \boldsymbol{x}_j \in \mathbb{R}^m$ is some observation data. The number of examples is denoted by $n$.

The log function is challenging, and for now we handle it by employing a basic gradient descent iteration (to solve the x-update sub-problem). The error is measured by comparing to the ground-truth, generated via CVX Grant & Boyd (2014).

### 4.2.1 BEYOND THE LIMIT

Here, we compare our operator stepsize with the underlying best scalar stepsize (the limit), which is found via grid search under a fixed random number generator. We consider 3 classes of data, each of 100 points/examples and $\mathbb{R}^{3\times 1}$ dimension.

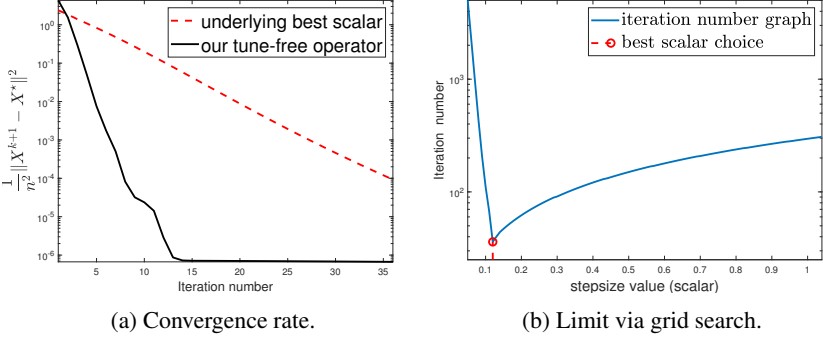

(a) Convergence rate.     (b) Limit via grid search.

Figure 3: Beyond the scalar-case limit, $m = 3, n = 100$.

*Remarks* 4.5. From Figure 3a, we observe that our algorithm converged at accuracy $10^{-6}$. This is due to the involvement of a log function, where the CVX employs an experimental successive estimation, and its solution (treated as the ground-truth) is of a low accuracy.

*Remarks* 4.6. From Figure 3b, we observe that the iteration number graph admits a very sharp curvature, implying high sensitivity to stepsize selection.

### 4.2.2 SCALABILITY

Here, we concern the scalability issue. We compare ours with two empirical scalar stepsize choices, 1 and 1.6, suggested in the milestone SDP paper Wen et al. (2010). We consider 3 classes of data, each of 1000 points/examples. We increase the dimension of the example to test the scalability issue.

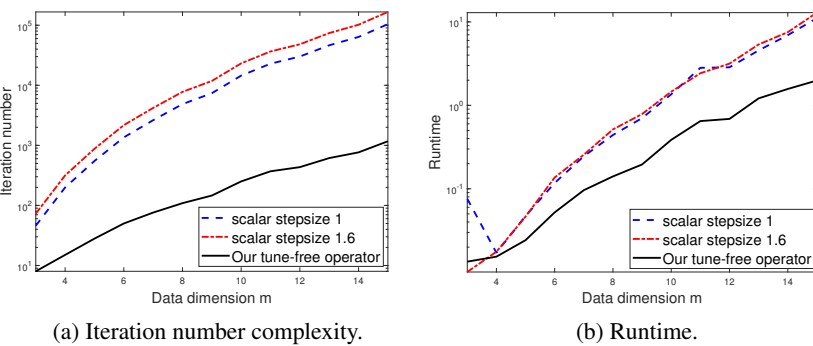

(a) Iteration number complexity.  (b) Runtime.

Figure 4: Scalability: our operator stepsize vs. fixed scalar 1 and 1.6.

*Remarks* 4.7. Figure 4a measures the iteration number complexity, and ours shows a significant advantage, roughly $10\times$ acceleration at the beginning stage and $100\times$ acceleration at the ending stage. Such an advantage appears consistently increasing with data dimension.

*Remarks* 4.8. Figure 4b measures the CPU runtime by a 'tic toc' command. Ours has a marginal disadvantage at the start, but soon gains advantage and arrives at roughly $10\times$ acceleration at the end. Our advantage is observed consistently increasing with data dimension.

## 5 CONCLUSION

For the first time, an operator stepsize is designed for semidefinite programming, with a special structure inspired by the Schur complement lemma. It enjoys several nice properties, including closed-form iterates, cheap computational cost (owing to dot product), and tune-free. Compared to the standard scalar stepsize, our operator one admits extra degrees of freedom, which mathematically allows it to surpass the acceleration limit (of the standard version). This aspect has been confirmed numerically, where preliminary tests show great advantages in iteration number complexity and runtime. Overall, we believe our operator ADMM significantly alleviated the long-standing scalability issue of semidefinite programming.

## 6 REPRODUCIBILITY STATEMENT

All figures in this manuscript can be reproduced, using MATLAB codes submitted as supplementary material.

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

# A APPENDIX

## A.1 PROOF OF LEMMA 2.1

Our Lemma 2.1, restated here as

**Lemma A.1** (guideline). *Given any invertible operator $\mathcal{S}$, with its inverse denoted as $\mathcal{S}^{-1}$. Suppose the following holds:*

$$\mathcal{S}^{-1}(\boldsymbol{Z}) \in \mathbb{S}_+^n, \qquad \forall \boldsymbol{Z} \in \mathbb{S}_+^n. \tag{A.1}$$

*Then, a closed-form solution is available:*

$$\mathcal{S}^{-1} \circ \Pi_{\mathbb{S}_+^n}(\mathcal{S}\boldsymbol{V}) = \underset{\boldsymbol{Z}}{argmin} \; \delta_{\mathbb{S}_+^n}(\boldsymbol{Z}) + \frac{1}{2}\|\boldsymbol{Z} - \boldsymbol{V}\|_{\mathcal{M}}^2,$$

$$= \underset{\boldsymbol{Z}}{argmin} \; \delta_{\mathbb{S}_+^n}(\boldsymbol{Z}) + \frac{1}{2}\|\mathcal{S}\boldsymbol{Z} - \mathcal{S}\boldsymbol{V}\|^2, \tag{A.2}$$

*where $\mathcal{M} = \mathcal{S} \circ \mathcal{S}$.*

*Proof.* equation A.1 implies

$$\delta_{\mathbb{S}_+^n}(\mathcal{S}^{-1}\bar{\boldsymbol{Z}}) = \delta_{\mathbb{S}_+^n}(\bar{\boldsymbol{Z}}). \tag{A.3}$$

From the right-hand side of equation A.2, we obtain

$$\underset{\boldsymbol{Z}}{argmin} \; \delta_{\mathbb{S}_+^n}(\boldsymbol{Z}) + \frac{1}{2}\|\mathcal{S}\boldsymbol{Z} - \mathcal{S}\boldsymbol{V}\|^2 = \mathcal{S}^{-1} \underset{\bar{\boldsymbol{Z}}}{argmin} \; \delta_{\mathbb{S}_+^n}(\mathcal{S}^{-1}\bar{\boldsymbol{Z}}) + \frac{1}{2}\|\bar{\boldsymbol{Z}} - \mathcal{S}\boldsymbol{V}\|^2,$$

$$= \mathcal{S}^{-1}\underset{\bar{\boldsymbol{Z}}}{argmin} \; \delta_{\mathbb{S}_+^n}(\bar{\boldsymbol{Z}}) + \frac{1}{2}\|\bar{\boldsymbol{Z}} - \mathcal{S}\boldsymbol{V}\|^2,$$

$$= \mathcal{S}^{-1} \circ \Pi_{\mathbb{S}_+^n}(\mathcal{S}\boldsymbol{V}). \tag{A.4}$$

where $\bar{\boldsymbol{Z}} = \mathcal{S}\boldsymbol{Z}$ is a variable substitution. The proof is now concluded. $\square$

## A.2 PROOF OF PROPOSITION 2.1

Our Proposition 2.1, restated here as

**Proposition A.1** (operator design). *Given scalars $\gamma_1, \gamma_2 > 0$ and any integer $m \in \{1, 2, \ldots, n-1\}$. Let operator $\mathcal{S}$ be defined as*

$$\mathcal{S}(\boldsymbol{V}) = \begin{bmatrix} \sqrt{\frac{\gamma_1}{\gamma_2}}\,\mathbf{1}_1 & \sqrt{\gamma_1}\,\mathbf{1}_0 \\ \sqrt{\gamma_1}\,\mathbf{1}_0 & \sqrt{\gamma_1\gamma_2}\,\mathbf{1}_2 \end{bmatrix} \odot \boldsymbol{V}, \tag{A.5}$$

*and its inverse being*

$$\mathcal{S}^{-1}(\boldsymbol{V}) = \begin{bmatrix} \sqrt{\frac{\gamma_2}{\gamma_1}}\,\mathbf{1}_1 & \frac{1}{\sqrt{\gamma_1}}\,\mathbf{1}_0 \\ \frac{1}{\sqrt{\gamma_1}}\,\mathbf{1}_0 & \frac{1}{\sqrt{\gamma_1\gamma_2}}\,\mathbf{1}_2 \end{bmatrix} \odot \boldsymbol{V}, \tag{A.6}$$

*where $\mathbf{1}_1 \in \mathbb{S}^m$, $\mathbf{1}_0 \in \mathbb{R}^{m \times (n-m)}$, $\mathbf{1}_2 \in \mathbb{S}^{n-m}$, and where $\mathbf{1}$ denotes the ones matrix (i.e., all entries being 1), by $\odot$ the element-wise multiplication.*

*Then,*

$$\mathcal{S}^{-1} \circ \Pi_{\mathbb{S}_+^n}(\mathcal{S}\boldsymbol{V}) = \underset{\boldsymbol{Z}}{argmin} \; \delta_{\mathbb{S}_+^n}(\boldsymbol{Z}) + \frac{1}{2}\|\mathcal{S}\boldsymbol{Z} - \mathcal{S}\boldsymbol{V}\|^2. \tag{A.7}$$

*Proof.* To start, we define a partition, specified by integer $m$:

$$\boldsymbol{X} = \begin{bmatrix} \boldsymbol{X}_1 & \boldsymbol{X}_0 \\ \boldsymbol{X}_0^T & \boldsymbol{X}_2 \end{bmatrix}, \tag{A.8}$$

where $\boldsymbol{X}_1 \in \mathbb{S}^m$, $\boldsymbol{X}_0 \in \mathbb{R}^{m \times (n-m)}$, $\boldsymbol{X}_2 \in \mathbb{S}^{n-m}$. By the generalized *Schur complement* argument, see (Boyd & Vandenberghe, 2004, A.5.5), we can rewrite $\mathcal{S}^{-1}(\boldsymbol{X})$ into

$$\begin{bmatrix} \sqrt{\frac{\gamma_2}{\gamma_1}}\,\boldsymbol{X}_1 & \frac{1}{\sqrt{\gamma_1}}\,\boldsymbol{X}_0 \\ \frac{1}{\sqrt{\gamma_1}}\,\boldsymbol{X}_0^T & \frac{1}{\sqrt{\gamma_1\gamma_2}}\,\boldsymbol{X}_2 \end{bmatrix} \succeq 0, \tag{A.9}$$

$$\iff \quad \sqrt{\frac{\gamma_2}{\gamma_1}}\,\boldsymbol{X}_1 \succeq 0, \quad \frac{1}{\sqrt{\gamma_1\gamma_2}}\,(\boldsymbol{X}_2 - \boldsymbol{X}_0^{\mathrm{T}}\boldsymbol{X}_1^{\dagger}\boldsymbol{X}_0) \succeq 0, \quad \frac{1}{\sqrt{\gamma_1}}\,(\boldsymbol{I} - \boldsymbol{X}_1\boldsymbol{X}_1^{\dagger})\boldsymbol{X}_0 = 0, \tag{A.10}$$

where $\cdot^{\dagger}$ denotes the pseudo-inverse. Due to $\gamma_1, \gamma_2$ related coefficients are all positive, the above holds if and only if $\boldsymbol{X} \succeq 0$. That is, $\mathcal{S}^{-1}(\boldsymbol{X}) \in \mathbb{S}^n_+$, $\forall \boldsymbol{X} \in \mathbb{S}^n_+$, i.e., equation 2.1 holds. Applying Lemma 2.1 then concludes the proof. $\qquad\square$

### A.3 PROOF THEOREM 3.1

Our Theorem 3.1, restated here as

**Theorem A.1.** *The worst-case optimal choice of stepsize $\mathcal{S}$, or equivalently the parameter pair $(\gamma_1, \gamma_2)$, is given by*

$$\gamma_1^{\star} = \sqrt{\frac{\gamma_2^{\star}\|\boldsymbol{\Lambda}_1^{\star}\|^2 + \frac{1}{\gamma_2^{\star}}\|\boldsymbol{\Lambda}_2^{\star}\|^2 + 2\|\boldsymbol{\Lambda}_0^{\star}\|^2}{\frac{1}{\gamma_2^{\star}}\|\hat{\boldsymbol{X}}_1^{\star}\|^2 + \gamma_2^{\star}\|\hat{\boldsymbol{X}}_2^{\star}\|^2 + 2\|\hat{\boldsymbol{X}}_0^{\star}\|^2}}. \tag{A.11}$$

*with $\gamma_2^{\star}$ being a positive root of the following degree-4 polynomial:*

$$\gamma_2^{\star 4}\|\hat{\boldsymbol{X}}_2^{\star}\|^2\|\boldsymbol{\Lambda}_1^{\star}\|^2 + \gamma_2^{\star 3}(\|\hat{\boldsymbol{X}}_2^{\star}\|^2\|\boldsymbol{\Lambda}_0^{\star}\|^2 + \|\hat{\boldsymbol{X}}_0^{\star}\|^2\|\boldsymbol{\Lambda}_1^{\star}\|^2) - \gamma_2^{\star}(\|\boldsymbol{\Lambda}_2^{\star}\|^2\|\hat{\boldsymbol{X}}_0^{\star}\|^2 + \|\boldsymbol{\Lambda}_0^{\star}\|^2\|\hat{\boldsymbol{X}}_1^{\star}\|^2)$$
$$- \|\boldsymbol{\Lambda}_2^{\star}\|^2\|\hat{\boldsymbol{X}}_1^{\star}\|^2 = 0, \tag{A.12}$$

*with a closed-form solution available.*

*Proof.* The minimizer is obtained when the derivative w.r.t. $\gamma_1$ and $\gamma_2$ vanishes, i.e.,

$$\frac{1}{\gamma_2^{\star}}\|\hat{\boldsymbol{X}}_1^{\star}\|^2 - \frac{\gamma_2^{\star}}{\gamma_1^{\star 2}}\|\boldsymbol{\Lambda}_1^{\star}\|^2 + \gamma_2^{\star}\|\hat{\boldsymbol{X}}_2^{\star}\|^2 - \frac{1}{\gamma_1^{\star 2}\gamma_2^{\star}}\|\boldsymbol{\Lambda}_2^{\star}\|^2 + 2\|\hat{\boldsymbol{X}}_0^{\star}\|^2 - \frac{2}{\gamma_1^{\star 2}}\|\boldsymbol{\Lambda}_0^{\star}\|^2 = 0, \tag{A.13}$$

$$-\frac{\gamma_1^{\star}}{\gamma_2^{\star 2}}\|\hat{\boldsymbol{X}}_1^{\star}\|^2 + \frac{1}{\gamma_1^{\star}}\|\boldsymbol{\Lambda}_1^{\star}\|^2 + \gamma_1^{\star}\|\hat{\boldsymbol{X}}_2^{\star}\|^2 - \frac{1}{\gamma_1^{\star}\gamma_2^{\star 2}}\|\boldsymbol{\Lambda}_2^{\star}\|^2 = 0, \tag{A.14}$$

By equation A.13, we instantly obtain the $\gamma_1^{\star}$ expression in our proposition. which can be rewritten into

$$\gamma_1^{\star 2}\|\hat{\boldsymbol{X}}_1^{\star}\|^2 - \gamma_2^{\star 2}\|\boldsymbol{\Lambda}_1^{\star}\|^2 + \gamma_1^{\star 2}\gamma_2^{\star 2}\|\hat{\boldsymbol{X}}_2^{\star}\|^2 - \|\boldsymbol{\Lambda}_2^{\star}\|^2 + 2\gamma_1^{\star 2}\gamma_2^{\star}\|\hat{\boldsymbol{X}}_0^{\star}\|^2 - 2\gamma_2^{\star}\|\boldsymbol{\Lambda}_0^{\star}\|^2 = 0, \tag{A.15}$$

$$-\gamma_1^{\star 2}\|\hat{\boldsymbol{X}}_1^{\star}\|^2 + \gamma_2^{\star 2}\|\boldsymbol{\Lambda}_1^{\star}\|^2 + \gamma_1^{\star 2}\gamma_2^{\star 2}\|\hat{\boldsymbol{X}}_2^{\star}\|^2 - \|\boldsymbol{\Lambda}_2^{\star}\|^2 = 0, \tag{A.16}$$

which, after simple manipulations, can be simplified to

$$\gamma_1^{\star 2}\gamma_2^{\star 2}\|\hat{\boldsymbol{X}}_2^{\star}\|^2 - \|\boldsymbol{\Lambda}_2^{\star}\|^2 + \gamma_1^{\star 2}\gamma_2^{\star}\|\hat{\boldsymbol{X}}_0^{\star}\|^2 - \gamma_2^{\star}\|\boldsymbol{\Lambda}_0^{\star}\|^2 = 0, \tag{A.17}$$

$$\gamma_1^{\star 2}\|\hat{\boldsymbol{X}}_1^{\star}\|^2 - \gamma_2^{\star 2}\|\boldsymbol{\Lambda}_1^{\star}\|^2 + \gamma_1^{\star 2}\gamma_2^{\star}\|\hat{\boldsymbol{X}}_0^{\star}\|^2 - \gamma_2^{\star}\|\boldsymbol{\Lambda}_0^{\star}\|^2 = 0. \tag{A.18}$$

Separating $\gamma_1^{\star}$ gives

$$\gamma_1^{\star 2} = \frac{\|\boldsymbol{\Lambda}_2^{\star}\|^2 + \gamma_2^{\star}\|\boldsymbol{\Lambda}_0^{\star}\|^2}{\gamma_2^{\star 2}\|\hat{\boldsymbol{X}}_2^{\star}\|^2 + \gamma_2^{\star}\|\hat{\boldsymbol{X}}_0^{\star}\|^2}, \quad \gamma_1^{\star 2} = \frac{\gamma_2^{\star 2}\|\boldsymbol{\Lambda}_1^{\star}\|^2 + \gamma_2^{\star}\|\boldsymbol{\Lambda}_0^{\star}\|^2}{\|\hat{\boldsymbol{X}}_1^{\star}\|^2 + \gamma_2^{\star}\|\hat{\boldsymbol{X}}_0^{\star}\|^2}, \tag{A.19}$$

which should hold simultaneously, yielding equation A.12. The other one equation A.11 follows instantly from equation A.13. The proof is now concluded. $\qquad\square$

## A.4 DISTANCE METRIC RESULT

The Figures below correspond to our Section 4.2.1, where we see that the learned metric simplifies the classification issue, see a detailed discussion of benefits from Xing et al. (2002b).

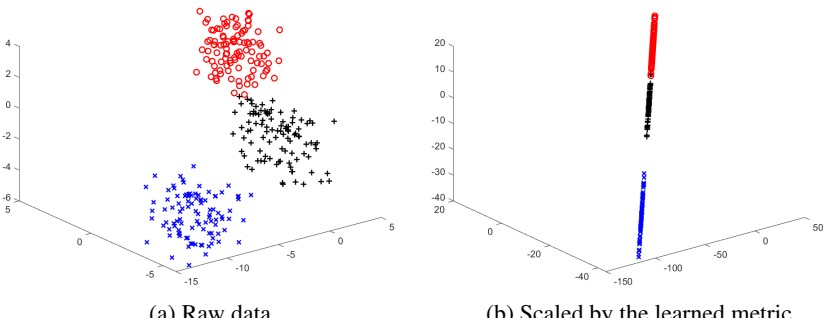

(a) Raw data.           (b) Scaled by the learned metric.

Figure 5: A visualized example: 3 classes of data.

