# Table of Contents

# Boolean least squares, Figure 1

This codes reproduce Figure 1 in our paper. Below, we compare ours with a fully-tuned standard ADMM. The full-tuning requires a grid search.

```matlab
clear;clc;
rng('default'); % random number generator
      n = 50;
    dim = n+1;
max_itr = 2e3;  % max iteration number
% data
     AA = randn(n);
     x0 = sign(randn(n,1)-0.5);
     b0 = AA*x0;
     A0 = [AA'*AA, AA'*b0; b0'*AA, 0];
% our operator ADMM
     Z = zeros(dim);
   Lam = zeros(dim);
     S = ones(dim);
for  i = 1 : max_itr
     X = Z - (Lam + A0)./S./S;
    X(1:dim+1:end) = 1;

 [V,D] = eig(S.*X + Lam./S);
     d = diag(D);
     d = max(d,0);
     Z = V*diag(d)*V'./S;

   Lam = Lam + S.*S.*(X - Z);
    % operator stepsize
     S = adapt_update(X, Lam, dim);
    % record
     X_ours{i} = X;
end

% ground-truth via CVX
cvx_begin sdp quiet
cvx_precision best
    variable  X(dim,dim) symmetric;
    minimize ( trace(A0'*X) )
    subject to
               X == semidefinite(dim);
         diag(X) == 1;
```

```matlab
cvx_end
X_star = X;

% primal iterates error
for j = 1:max_itr
error_ours(j) = norm(X_ours{j} - X_star,'fro')^2/dim^2;
end
```

# best scalar via grid search

```matlab
thre = 1e-10; %error threshold
for i = 1 : 100
        gamma = 10 + (i-1) * 0.3;
[itr, error] = ADMM_scalar(gamma,dim,A0,X_star,thre,max_itr);

  itr_rec(i) = itr;  % iteration number
gamma_rec(i) = gamma;% stepsize value
error_rec{i} = error;% error
end
% the fully-tuned scalar
[min_itr,index] = min(itr_rec);
     gamma_best = gamma_rec(index);
     error_best = error_rec{index};
% Figure 1a
figure
semilogy(error_best,'--r','LineWidth',2);hold on;
semilogy(error_ours(1:length(error_best)),'-k','LineWidth',2);
axis tight;

xlabel('Iteration number','FontSize',14);
ylabel('$$ \frac{1}{n^2}\Vert X^{k+1} - X^\star \Vert^2$$',...
        'Interpreter','latex','FontSize',18);
legend('underlying best scalar',...
        'our tune-free operator','FontSize',20);
% Figure 1b
figure
semilogy(gamma_rec,itr_rec,'LineWidth',2,...
          'color','[0.00,0.45,0.74]');hold on;
stem(gamma_best,min_itr,'--r','Marker','o',...
     'markersize',8,'LineWidth',2);

xlabel('stepsize value (scalar)','FontSize',16);
ylabel('Iteration  number','FontSize',14);
legend('iteration number graph','best scalar choice',...
        'interpreter','latex','FontSize',20);
axis tight;
```

# scalar ADMM, function

```matlab
function [itr, error] = ADMM_scalar(gamma,dim,...
                        A0,X_star,err_thre,max_itr)
        Z = zeros(dim);
```

```matlab
    Lam = zeros(dim);
for itr = 1:max_itr
    X = Z - (Lam + A0)/gamma;
    X(1:dim+1:end) = 1;

  [V,D] = eig(X + Lam/gamma);
    d = diag(D);
    d = max(d,0);
    Z = V*diag(d)*V';

    Lam = Lam + gamma*(X - Z);
    % error
    error(itr) = norm(X - X_star,'fro')^2/dim^2;
    if error(itr) < err_thre
        break;
    end
end
end
```

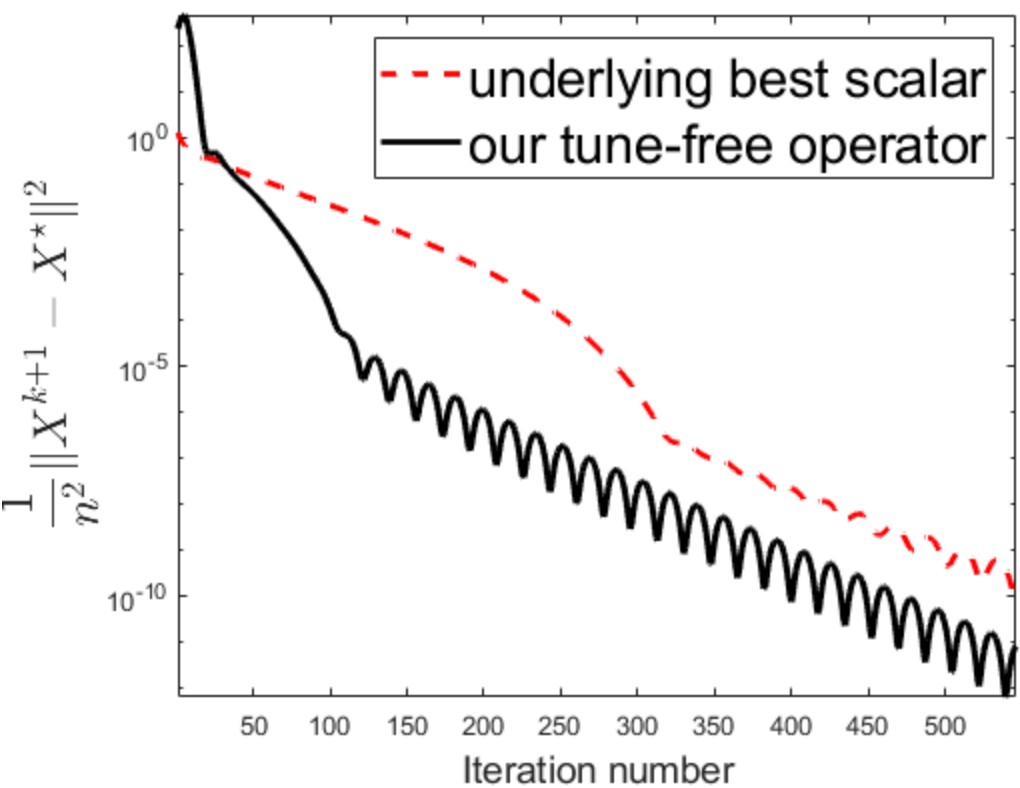

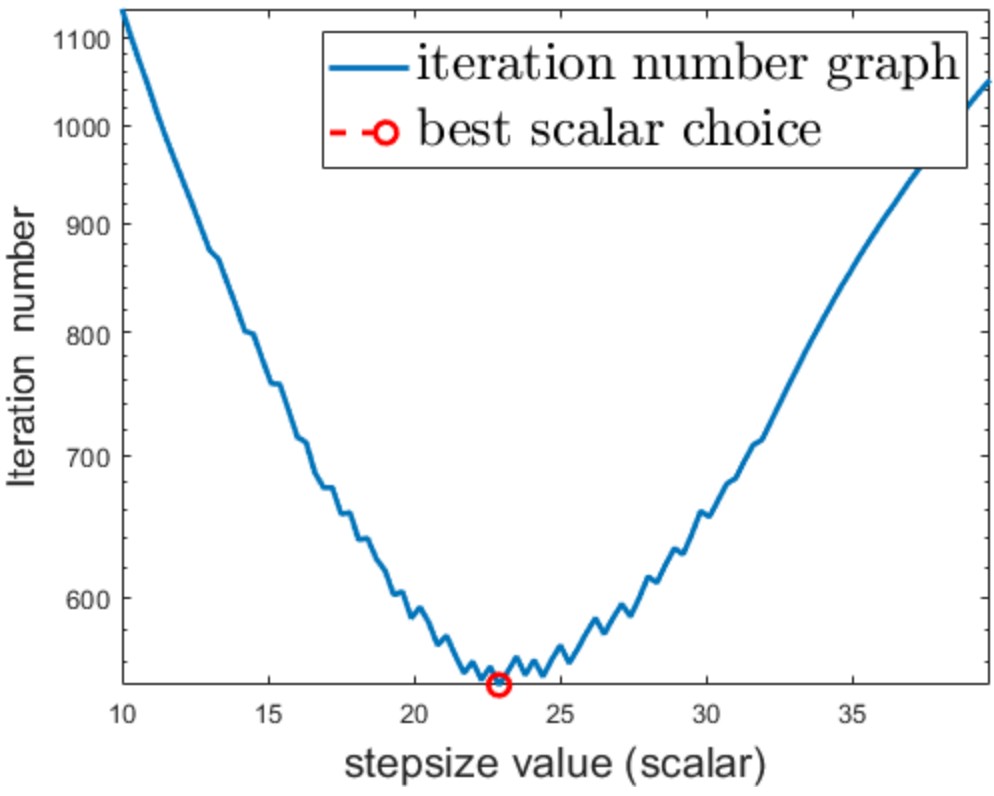

# Our degree-4 polynomial, function

```
function S = adapt_update (X, Lambda, dim)
    j0 = dim - 1;          % default partition choice
     S = ones(dim);        % operator stepsize
    % partitioning
    X1 = X(1:j0,1:j0);
    X2 = X(j0+1:end,j0+1:end);
    X0 = X(j0+1:end,1:j0);
    Lambda1 = Lambda(1:j0,1:j0);
    Lambda2 = Lambda(j0+1:end,j0+1:end);
    Lambda0 = Lambda(j0+1:end,1:j0);
    % coefficients
    a =   norm(X2,'fro')^2 * norm(Lambda1,'fro')^2 ;
    b =   norm(X2,'fro')^2 * norm(Lambda0,'fro')^2 ...
        + norm(X0,'fro')^2 * norm(Lambda1,'fro')^2;
    d = - norm(X0,'fro')^2 * norm(Lambda2,'fro')^2 ...
        - norm(X1,'fro')^2 * norm(Lambda0,'fro')^2;
    e = - norm(X1,'fro')^2 * norm(Lambda2,'fro')^2;
    % closed-form solutions
    p1 = sqrt(27)/2*(a*d^2 + b^2 * e);
    p2 = p1 + sqrt((b*d - 4*a*e)^3 + p1^2);
    % compute p3
    input  = p2;
    radius = abs(input);
```

```matlab
    angle  = real(input/radius);
    temp   = acos(angle)/3;
    root0  = nthroot(radius,3)*(cos(temp) + 1i*sin(temp));

    p3 = ((-b*d + 4*a*e)/root0 + root0)/a/sqrt(3);
    p4 = sqrt(b^2/4/a^2+p3);
    p5 = b^2/2/a^2 - p3;
    p6 = (-b^3/a^3 - 8*d/a)/4/p4;
    % 4 roots of the degree-4 polynomial
    root(1) = -b/4/a - p4/2 - sqrt(p5 - p6)/2;
    root(2) = -b/4/a - p4/2 + sqrt(p5 - p6)/2;
    root(3) = -b/4/a + p4/2 - sqrt(p5 + p6)/2;
    root(4) = -b/4/a + p4/2 + sqrt(p5 + p6)/2;
    % select a positive real choice
    gamma2 = root( real(root)>0 & abs(imag(root)) < 1e-1);
    gamma2 = real(gamma2(1));
    % compute gamma1
    gamma1 = norm(Lambda./S,'fro')/norm(S.*X,'fro');
    % compute the operator stepsize
    S(1:j0,1:j0)         = 1/sqrt(gamma2);
    S(j0+1:end,j0+1:end) = sqrt(gamma2);
    S = sqrt(gamma1)*S;
end
```

*Published with MATLAB® R2020b*

# Boolean least squares, Figure 2

This codes reproduce Figure 2 in our paper. Below, we test the scalability performance of our solver and standard scalar ADMM. CVX is used to generate ground-truth, and all algorithms will stop when a certain error threshold attained.

```matlab
clear;clc;
rng('default')
for    out = 1 : 11
         n = 10 + (out-1)*10;
       dim = n+1;
   max_itr = 1e4;
error_thre = 1e-4;
% data
AA = randn(n);
x0 = sign(randn(n,1)-0.5);
b0 = AA*x0;
A0 = [AA'*AA, AA'*b0; b0'*AA, 0 ];
% CVX: generate ground-truth to measure error
cvx_begin sdp quiet
cvx_precision best
    variable  X(dim,dim) symmetric;
    dual variable Lam_cvx
    minimize ( trace(A0'*X) )
    subject to
    Lam_cvx:       X == semidefinite(dim);
             diag(X) == 1;
cvx_end
X_cvx = X;

% (i) our operator ADMM
tic
  Z = zeros(dim);
Lam = zeros(dim);
  S = ones(dim);
for i_ours = 1:max_itr
    % primal update
      X = Z - (Lam + A0)./S./S;
    X(1:n+2:end) = 1;
    % z-update
  [V,D] = eig(S.*X + Lam./S);
      d = diag(D);
      d = max(d,0);
      Z = V*diag(d)*V'./S;
    % dual update
    Lam = Lam + S.*S.*(X - Z);
    % operator stepsize with early stop
    if i_ours > 1 && primal_error > 0.1
    S = adapt_update(X, Lam, dim);
    end
    % stopping criteria
       primal_error = norm(X - X_cvx,'fro')^2/dim^2;
```

```matlab
        if primal_error <= error_thre
            break;
        end
    end
end
iter_ours(out) = i_ours;
time_ours(out) = toc;
% (ii) ADMM: scalar stepsize 1
tic
gamma = 1;
    Z = zeros(dim);
  Lam = zeros(dim);
for i_scalar = 1:max_itr
    % primal update
    X = Z - (Lam + A0)/gamma;
    X(1:dim+1:end) = 1;
    % z-update
    [V,D] = eig(X + Lam/gamma);
    d = diag(D);
    d = max(d,0);
    Z = V*diag(d)*V';
    % dual update
    Lam = Lam + gamma*(X - Z);
    % stopping criteria
    primal_error = norm(X - X_cvx,'fro')^2/dim^2;
    if primal_error <= error_thre
        break;
    end
end
iter_scalar(out) = i_scalar;
time_scalar(out) = toc;

% (iii) ADMM: scalar stepsize 1.6
tic
gamma = 1.6;
    Z = zeros(dim);
  Lam = zeros(dim);
for i_scalar2 = 1:max_itr
    % primal update
    X = Z - (Lam + A0)/gamma;
    X(1:dim+1:end) = 1;
    % z-update
    [V,D] = eig(X + Lam/gamma);
    d = diag(D);
    d = max(d,0);
    Z = V*diag(d)*V';
    % dual update
    Lam = Lam + gamma*(X - Z);
    % stopping criteria
    primal_error = norm(X - X_cvx,'fro')^2/dim^2;
    if primal_error <= error_thre
        break;
    end
end
iter_scalar2(out) = i_scalar2;
```

```matlab
        time_scalar2(out) = toc;
    end
    % larger data size
    for out = 12 : 40
         n = 10 + (out-1)*10;
        dim = n+1;
max_itr = 1e4;
% data
AA = randn(n);
x0 = sign(randn(n,1)-0.5);
b0 = AA*x0;
A0 = [AA'*AA, AA'*b0; b0'*AA, 0 ];
% CVX: generate ground-truth to measure error
cvx_begin sdp quiet
cvx_precision best
    variable      X(dim,dim) symmetric;
    dual variable Lam_cvx
    minimize ( trace(A0'*X) )
    subject to
    Lam_cvx:      X == semidefinite(dim);
             diag(X) == ones(dim,1);
cvx_end
X_cvx = X;
% (iv) our operator ADMM
tic
    Z = zeros(dim);
  Lam = zeros(dim);
    S = ones(dim);
for i_ours = 1:max_itr
    % primal update
     X = Z - (Lam + A0)./S./S;
    X(1:n+2:end) = 1;
    % z-update
 [V,D] = eig(S.*X + Lam./S);
     d = diag(D);
     d = max(d,0);
     Z = V*diag(d)*V'./S;
    % dual update
    Lam = Lam + S.*S.*(X - Z);
    % operator stepsize update
    if i_ours > 1 &&  primal_error > 0.1
     S = adapt_update(X, Lam, dim);
    end
    % stopping criteria
    primal_error = norm(X - X_cvx,'fro')^2/dim^2;
    if primal_error <= error_thre
        break;
    end
end
iter_ours(out) = i_ours;
time_ours(out) = toc;
end

% Figure 2a
```

```matlab
figure
semilogy(10:10:1.1e2,iter_scalar,'--b','LineWidth',2);hold on;
semilogy(10:10:1.1e2,iter_scalar2,'-.r','LineWidth',2);
semilogy(10:10:n,iter_ours,'-k','LineWidth',2);
axis tight;

ylabel('Iteration number','FontSize',16);
xlabel('Data size','FontSize',16);
legend('scalar stepsize 1',...
'scalar stepsize 1.6',...
'Our tune-free operator','FontSize',20);

% Figure 2b
figure
semilogy(10:10:1.1e2,time_scalar,'--b','LineWidth',2);hold on;
semilogy(10:10:1.1e2,time_scalar2,'-.r','LineWidth',2);
semilogy(10:10:n,time_ours,'-k','LineWidth',2);
axis tight;

ylabel('Runtime','FontSize',16);
xlabel('Data size','FontSize',16);
legend('scalar stepsize 1',...
'scalar stepsize 1.6',...
'Our tune-free operator','FontSize',20);
```

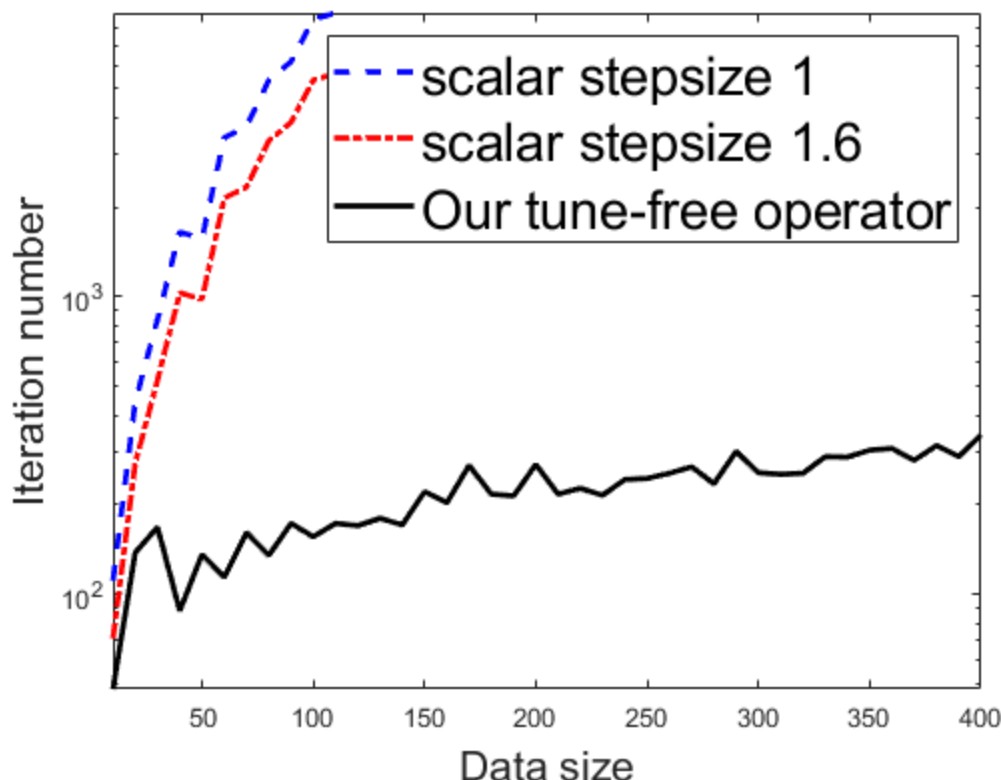

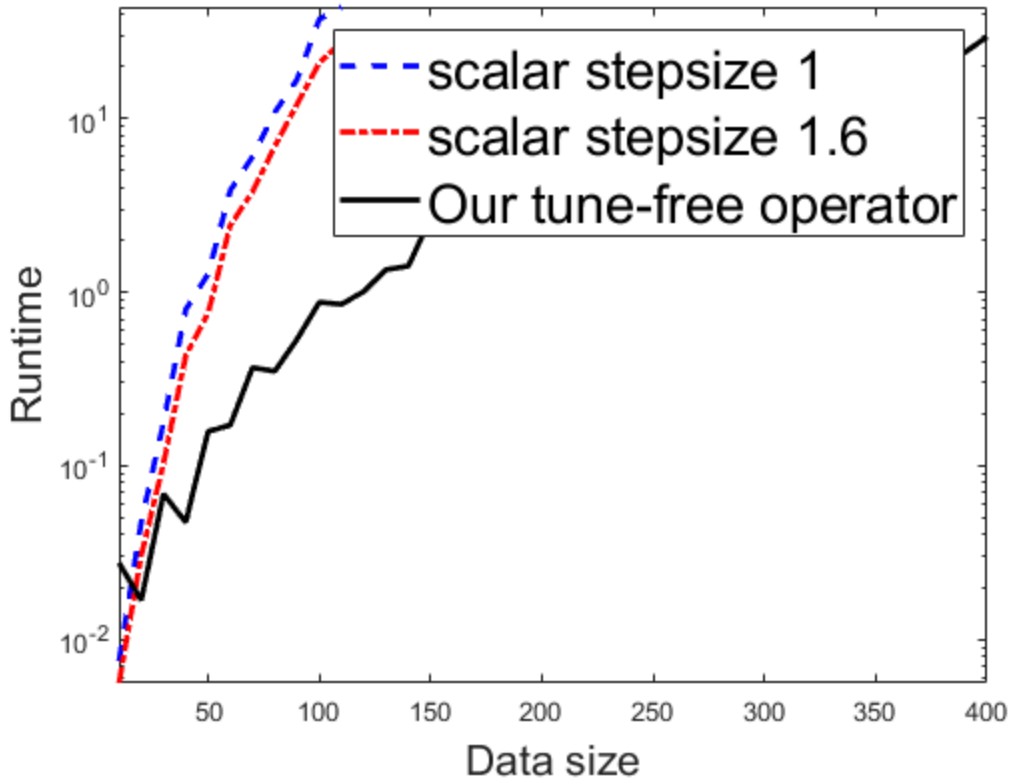

*Published with MATLAB® R2020b*

## Table of Contents

# Distance metric learning (beyond the limit)

This code reproduces our Figure 3.

```
clear;clc;
rng('default');         % random number generator
      k = 3;            % classes of data
      n = 1e2;          % number of data points
max_itr = 5e3;          % max iteration number
   thre = 1e-4;         % required accuracy
% synthetic data generation
    b1 = rand(k,n)+ randi([0,3],k,n);
    b2 = rand(k,n)+ randi([-3,0],k,n)...
         + [0;1;zeros(k-2,1)] * randi([-3,-1],1,n);
    b3 = rand(k,n)+ randi([-6,-3],k,n)...
         + [1;0;zeros(k-2,1)] * randi([-8,-5],1,n);
    nn = length(b1);

    B1 = []; B2 = []; B3 = [];
for  i = 1:nn
  tmp1 = b1(:,i)*ones(nn-i,1)' - b1(:,i+1:end);
    B1 = [B1,tmp1];
  tmp2 = b1(:,i)*ones(nn,1)' - b2;
    B2 = [B2,tmp2];
  tmp3 = b1(:,i)*ones(nn,1)' - b3;
    B3 = [B3,tmp3];
end
   BB1 = B1*B1' / nn^2/k^2;
   BB2 = B2*B2' / nn^2/k^2 + B3*B3' / nn^2/k^2;

% initialization
 X_tmp = 1e-4*randn(k);
 X_ini = X_tmp'*X_tmp;
% our operator ADMM
     X = X_ini;
     Z = zeros(k);
   Lam = zeros(k);
     S = ones(k);
for  i = 1 : max_itr
    % x-update
    for  j = 1:1e3
        dX = BB1 - 1/trace(X*BB2)*BB2 + S.*S.*(X - Z) + Lam;
         X = X - 0.1*dX;
         if norm(dX,'fro') <= 1e-4
             break;
```

```matlab
            end
        end
        % z-update
        T = S.*X + Lam./S;
 [V,D] = eig(T/2 + T'/2);
        d = diag(D);
        d = max(d,0);
        Z = V*diag(d)*V'./S;
    % lambda-update
    Lam = Lam + S.*S.*(X - Z);
    % operator stepsize update
        S = adapt_update(X, Lam, k);
    % recording
        X_ours{i} = X;
    end
```

# Ground-truth error computation

```matlab
% ground-truth via the cvx package
cvx_begin quiet
cvx_precision best
        variables A(k,k)
        minimize (trace(A*BB1) - log(trace(A*BB2))) ;
        subject to
          A == semidefinite(k);
cvx_end
  X_star = A;
% compute our error
for    j = 1:max_itr
error_ours(j) = norm(X_ours{j} - X_star,'fro')^2/k^2;
end

% limit: search best scalar stepsize
for     i = 1 : 1e2
        gamma = 0.05 + (i-1)*0.01;
[itr, error] =
 ADMM_scalar_thre(gamma,k,BB1,BB2,X_ini,X_star,thre,max_itr);

  itr_rec(i) = itr;  % iteration number
gamma_rec(i) = gamma;% stepsize value
error_rec{i} = error;% error
end
% the fully-tuned scalar
[min_itr,index] = min(itr_rec);
     gamma_best = gamma_rec(index);
     error_best = error_rec{index};

% Figure 1a
grid_len = length(error_best);
figure
semilogy(error_best,'--r','LineWidth',2);hold on;
semilogy(error_ours(1:grid_len),'-k','LineWidth',2);
```

```matlab
xlabel('Iteration number','FontSize',14);
ylabel('$$ \frac{1}{n^2}\Vert X^{k+1} - X^\star \Vert^2$$',...
        'Interpreter','latex','FontSize',18);
legend('underlying best scalar',...
        'our tune-free operator','FontSize',20);
axis tight;

% Figure 1b
figure
semilogy(gamma_rec,itr_rec,'LineWidth',2,...
          'color','[0.00,0.45,0.74]');hold on;
    stem(gamma_best,min_itr,'--r','Marker','o',...
          'markersize',8,'LineWidth',2);

xlabel('stepsize value (scalar)','FontSize',16);
ylabel('Iteration  number','FontSize',14);
legend('iteration number graph','best scalar choice',...
        'interpreter','latex','FontSize',20);
axis tight;
```

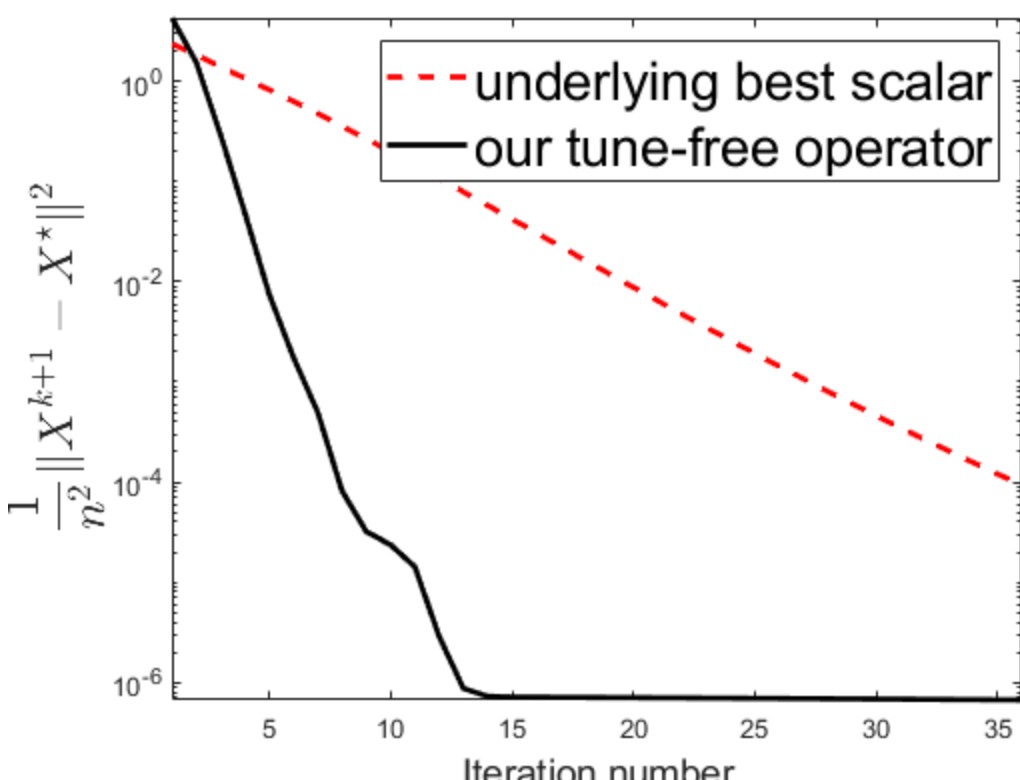

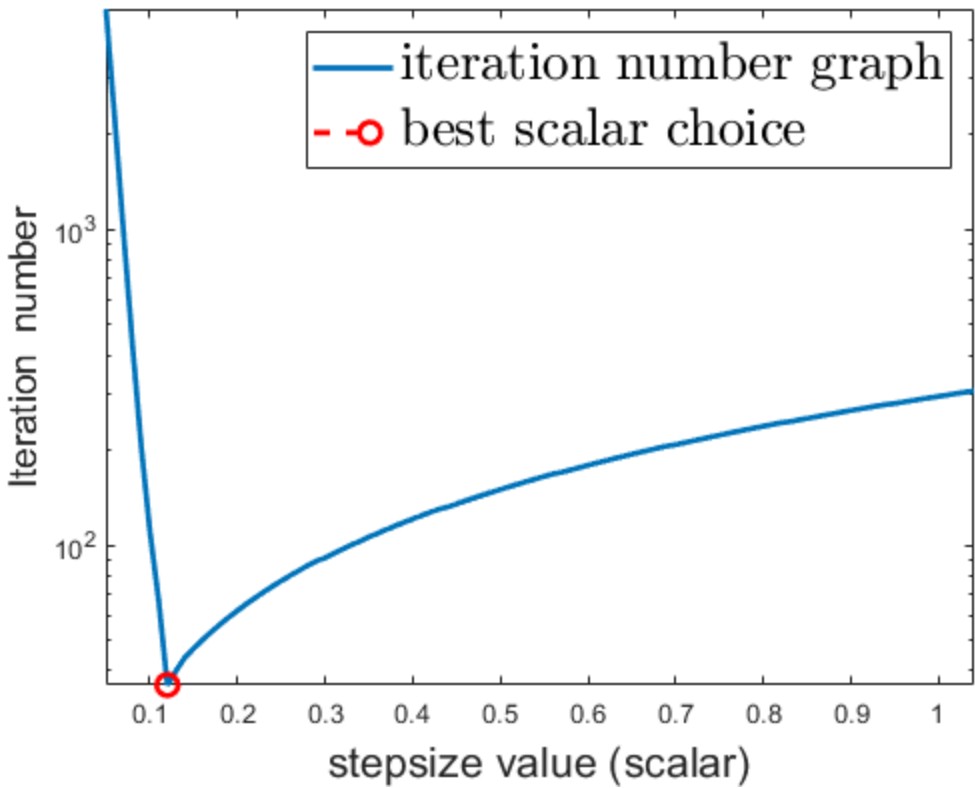

## visualize data

```
% 1. original
figure
scatter3(b1(1,:),b1(2,:),b1(3,:),'or','LineWidth',1.3);hold on;
scatter3(b2(1,:),b2(2,:),b2(3,:),'+k','LineWidth',1.3);
scatter3(b3(1,:),b3(2,:),b3(3,:),'xb','LineWidth',1.3);
grid off

% 2. metric scaled
b1_s = X*b1;
b2_s = X*b2;
b3_s = X*b3;

figure
scatter3(b1_s(1,:),b1_s(2,:),b1_s(3,:),'or','LineWidth',1.3);hold on;
scatter3(b2_s(1,:),b2_s(2,:),b2_s(3,:),'+k','LineWidth',1.3);
scatter3(b3_s(1,:),b3_s(2,:),b3_s(3,:),'xb','LineWidth',1.3);
grid off
```

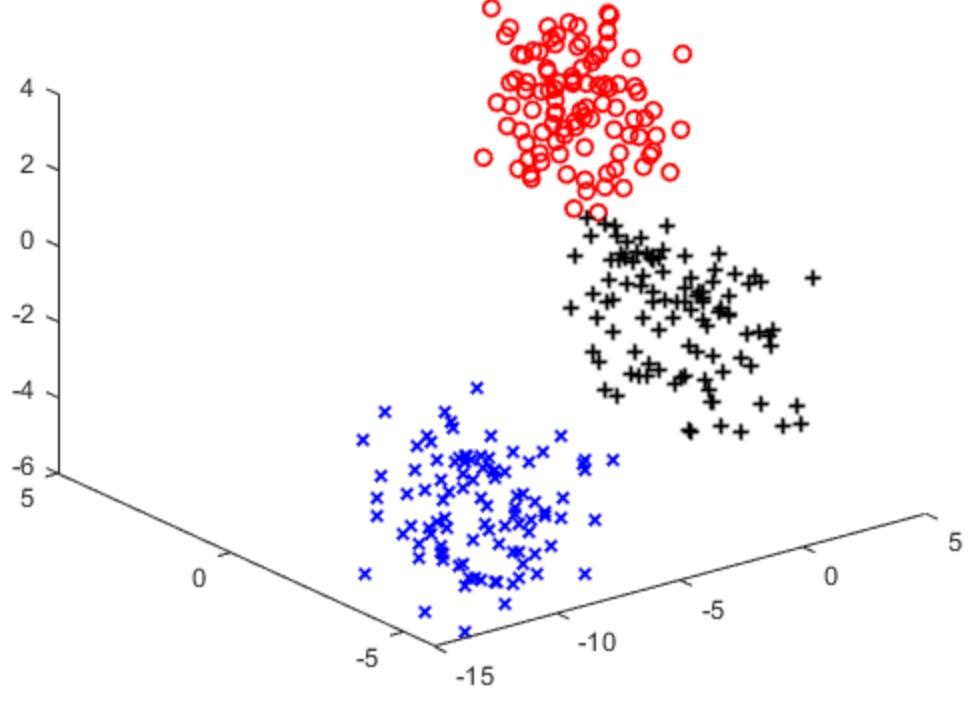

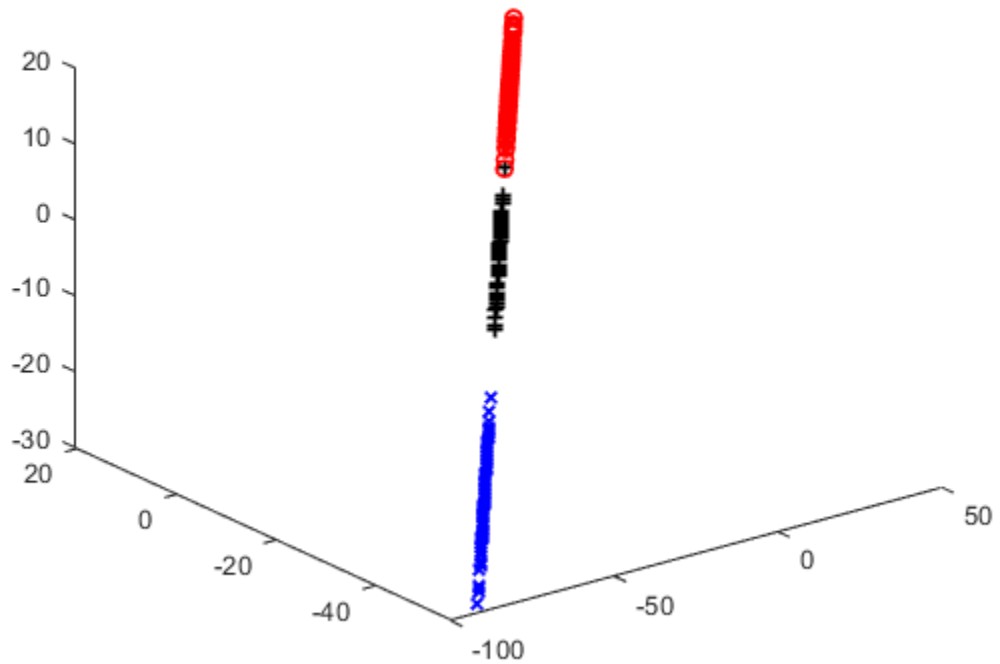

# our degree-4 polynomial

```matlab
function S = adapt_update (X, Lambda, dim)
    j0 = dim-1;          % default partition choice
     S = ones(dim);      % operator stepsize
    % partitioning
    X1 = X(1:j0,1:j0);
    X2 = X(j0+1:end,j0+1:end);
    X0 = X(j0+1:end,1:j0);
    Lambda1 = Lambda(1:j0,1:j0);
    Lambda2 = Lambda(j0+1:end,j0+1:end);
    Lambda0 = Lambda(j0+1:end,1:j0);
    % coefficients
    a =   norm(X2,'fro')^2 * norm(Lambda1,'fro')^2 ;
    b =   norm(X2,'fro')^2 * norm(Lambda0,'fro')^2 ...
        + norm(X0,'fro')^2 * norm(Lambda1,'fro')^2;
    d = - norm(X0,'fro')^2 * norm(Lambda2,'fro')^2 ...
        - norm(X1,'fro')^2 * norm(Lambda0,'fro')^2;
    e = - norm(X1,'fro')^2 * norm(Lambda2,'fro')^2;
    % closed-form solutions
    p1 = sqrt(27)/2*(a*d^2 + b^2*e);
    p2 = p1 + sqrt((b*d - 4*a*e)^3 + p1^2);

    radius = abs(p2);
    angle  = real(p2/radius);
    temp   = acos(angle)/3;
    root0  = nthroot(radius,3)*(cos(temp) + 1i*sin(temp));

    p3 = ((-b*d + 4*a*e)/root0 + root0)/a/sqrt(3);
    p4 = sqrt(b^2/4/a^2+p3);
    p5 = b^2/2/a^2 - p3;
    p6 = (-b^3/a^3 - 8*d/a)/4/p4;
    % 4 roots of the degree-4 polynomial
    root(1) = -b/4/a - p4/2 - sqrt(p5 - p6)/2;
    root(2) = -b/4/a - p4/2 + sqrt(p5 - p6)/2;
    root(3) = -b/4/a + p4/2 - sqrt(p5 + p6)/2;
    root(4) = -b/4/a + p4/2 + sqrt(p5 + p6)/2;
    % select a positive real choice
    gamma2 = root(real(root)>0 & abs(imag(root)) < 1e-1);
    gamma2 = real(gamma2(1));
    % compute gamma1
    gamma1 = norm(Lambda./S,'fro')/norm(S.*X,'fro');
    % compute the operator stepsize
    S(1:j0,1:j0)        = 1/sqrt(gamma2);
    S(j0+1:end,j0+1:end) = sqrt(gamma2);
    S = sqrt(gamma1)*S;
end
```

*Published with MATLAB® R2020b*

# Distance metric learning (scalability)

This code reproduces our Figure 4.

```matlab
clear;clc;
   thre = 1e-1;            % required accuracy
max_itr = 1e6;             % max iteration number
for  ii = 1: 13
rng('default');            % random number generator
     m = 3 + (ii-1);       % data dimension
     n = 1e3;              % data points number
% synthetic data generation
   b1 = rand(m,n)+ randi([0,3],m,n);
   b2 = rand(m,n)+ randi([-3,0],m,n)...
        + [0;1;zeros(m-2,1)] * randi([-3,-1],1,n);
   b3 = rand(m,n)+ randi([-6,-3],m,n)...
        + [1;0;zeros(m-2,1)] * randi([-8,-5],1,n);
   nn = length(b1);

   B1 = []; B2 = []; B3 = [];
for  i = 1:nn
  tmp1 = b1(:,i)*ones(nn-i,1)' - b1(:,i+1:end);
    B1 = [B1,tmp1];
  tmp2 = b1(:,i)*ones(nn,1)' - b2;
    B2 = [B2,tmp2];
  tmp3 = b1(:,i)*ones(nn,1)' - b3;
    B3 = [B3,tmp3];
end
   BB1 = B1*B1' / nn^2/m^2;
   BB2 = B2*B2' / nn^2/m^2 + B3*B3' / nn^2/m^2;

% ground-truth via CVX
cvx_begin quiet
cvx_precision best
      variables A(m,m)
      minimize (trace(A*BB1) - log(trace(A*BB2))) ;
      subject to
        A == semidefinite(m);
cvx_end
X_star = A;

% initialization
 X_tmp = 1e-4*randn(m);
 X_ini = X_tmp'*X_tmp;

% our operator ADMM
tic
    X = X_ini;
    Z = zeros(m);
  Lam = zeros(m);
    S = ones(m);
for  i = 1 : max_itr
```

```matlab
        % x-update
        for  j = 1:1e3
            dX = BB1 - 1/trace(X*BB2)*BB2 + S.*S.*(X - Z) + Lam;
             X = X - 0.1*dX;
              if norm(dX,'fro') <= 1e-5
                    break;
              end
        end
        % z-update
             T = S.*X + Lam./S;
        [V,D] = eig(T/2 + T'/2);
             d = diag(D);
             d = max(d,0);
             Z = V*diag(d)*V'./S;
        % lambda-update
         Lam = Lam + S.*S.*(X - Z);
        % early stop (save runtime)
        if j < 1e2
            S = adapt_update(X, Lam, m);
        end
        error_ours = norm(X - X_star,'fro')^2/m^2;
        if error_ours <= thre
             break
        end
    end
 itr_ours(ii) = i;
time_ours(ii) = toc;

% scalar stepsize 1
tic
        gamma = 1;
[itr, error] =
 ADMM_scalar_thre(gamma,m,BB1,BB2,X_ini,X_star,thre,max_itr);
  time_1(ii) = toc;
   itr_1(ii) = itr;  % iteration number
% scalar stepsize 1.6
tic
        gamma = 1.6;
[itr, error] =
 ADMM_scalar_thre(gamma,m,BB1,BB2,X_ini,X_star,thre,max_itr);
  time_2(ii) = toc;
   itr_2(ii) = itr;  % iteration number
end

% Figures
grid = 3 : m;
figure
semilogy(grid,itr_1,'--b','LineWidth',2);hold on;
semilogy(grid,itr_2,'-.r','LineWidth',2);
semilogy(grid,itr_ours,'-k','LineWidth',2);
axis tight;

ylabel('Iteration number','FontSize',16);
xlabel('Data dimension m','FontSize',16);
```

```matlab
legend('scalar stepsize 1',...
'scalar stepsize 1.6',...
'Our tune-free operator','FontSize',16);

figure
semilogy(grid,time_1,'--b','LineWidth',2);hold on;
semilogy(grid,time_2,'-.r','LineWidth',2);
semilogy(grid,time_ours,'-k','LineWidth',2);
axis tight;

ylabel('Runtime','FontSize',16);
xlabel('Data dimension m','FontSize',16);
legend('scalar stepsize 1',...
'scalar stepsize 1.6',...
'Our tune-free operator','FontSize',16);
```

# function: scalar ADMM

```matlab
function [i, error] =
 ADMM_scalar_thre(gamma,k,BB1,BB2,X_ini,X_star,thre,max_itr)
    X = X_ini;
    Z = zeros(k);
  Lam = zeros(k);
for i = 1 : max_itr
    for j = 1:1e3
       dX = BB1 - 1/trace(X*BB2)*BB2 + gamma * (X - Z) + Lam;
        X = X - 0.1*dX;
        if norm(dX,'fro') <= 1e-5
            break;
        end
    end
    % z-update
    T = X + Lam/gamma;
[V,D] = eig(T/2 + T'/2);
    d = diag(D);
    d = max(d,0);
    Z = V*diag(d)*V';
    % dual update
  Lam = Lam + gamma*(X - Z);
      error(i) = norm(X - X_star,'fro')^2/k^2;
    if error(i) <= thre
        break;
    end
end
end
```

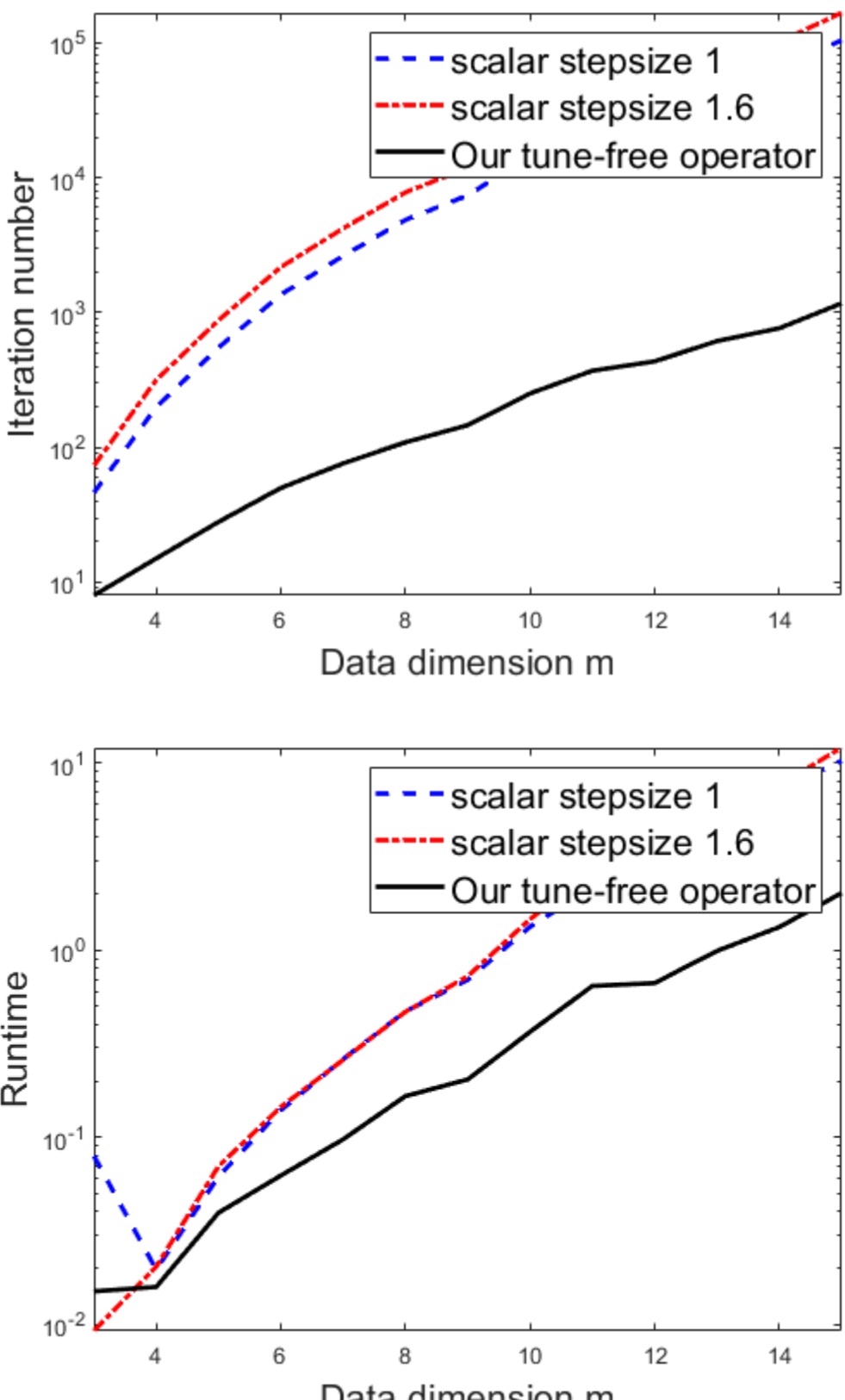

*Published with MATLAB® R2020b*