# OpenReview forum: "Accelerating semidefinite programming beyond limit: ADMM with tune-free operator stepsize"
_ICLR.cc/2025/Conference — Submitted to ICLR 2025_

### Official Review · Reviewer_p4sJ · 2024-11-01

**Soundness:** 2
**Presentation:** 2
**Contribution:** 2
**Rating:** 3
**Confidence:** 4

**Summary:**

This paper develops an operator stepsize for ADMM in the SDP context. The operator stepsize leads to a natural choice of algorithm parameters. Numerical experiments demonstrate the empirical performance of the proposed approach.

**Strengths:**

**Strength**

The paper is, in general, well-written, with some experiments demonstrating the performance of the proposed method.

**Weaknesses:**

**Weaknesses**

Although the paper claims it alleviates the scalability issues of SDP, I feel it's an overclaim after reading through the paper. In particular,

1. Remaining scalability issues

   The authors mention that the scalability issues of SDP originate from the large data dimension. This is generally true since the computational bottlenecks for SDP algorithms are basically

   - evaluating the barrier Hessian and scaling matrix (for IPM)
   - solving for the search direction (solving the augmented system or the reduced Schur complement system)
   - ensuring positive-definiteness of the conic variables (orthogonal projection onto $\mathbb{S}^n_+$or performing a ratio test).

   Among these, orthgonal projection and solving for the search direction are typical bottleneck routines for non-IPM-based methods.  However, according to the description of the paper, the proposed algorithm still relies on projection and linear system solving at each iteration. I'm not convinced that this addresses the scalability issues of SDP if it only saves the number of iterations of ADMM. When the number of constraints and SDP cone dimension becomes large, it may not be possible for the proposed method to complete a single iteration.

2. Fairness of comparison

   The paper mentions in several places that it goes "beyond limit". This mainly refers to taking a matrix stepsize in the dual update without increasing the computational cost that much. The authors show that their matrix stepsize outperforms the best possible single fixed stepsize [4]. However, operator stepsize used in the paper is essentially adaptive: it changes in different iterations. However, the benchmark is some optimal non-adaptive scalar stepsize. Given the popular adaptive heuristics for choosing $\rho$ in the ADMM literature [1], I do not think it is a fair comparison.

3. Insufficient experiments

   The paper seems to provide a general methodology to improve ADMM for SDP solving, but the tested benchmark is still restricted to relatively simple synthetic SDP problems of moderate dimension. This does not validate "alleviating the scalability issue of ADMM" as the authors claimed. I would suggest the authors perform experiments on problems with both large constraint number and cone dimension.

4. Presentation issues

   The paper contains several inconsistencies and presentation issues (See minor issues). These make the paper less accessible to the readers.

Given the limitations above, overall I feel the paper cannot be published in its current form at ICLR.

**Questions:**

**Questions**

1. According to the definition of $\mathbf{1}$ (all-one matrix), the matrix $\mathcal{S}$ has a principle submatrix that is not positive definite (e.g. equation 2.3, the first $m \times m$ submatrix is all one and not positive definite). How can its inverse be computed? Does the convergence guarantee of ADMM still hold?
2. Regarding the experiments, does the proposed approach really scale when the cone dimension and the number of constraints get large? Could you give additional experiments on real large-scale SDP instances?

**Minor issues**

1. Line 32.

   Ellipsoid method is the first polynomial-time algorithm for LP.

2. Line 51

   The review of typical first-order methods for SDP is not comprehensive. Methods like spectral bundle method and Frank-Wolfe are not mentioned. Please provide a more comprehensive review of first-order methods for SDP [3].

3. Line 81

   A solution is assumed exists => A solution is assumed to exist.

4. Line 102

   SDP include => SDP includes.

5. Line 112

   The sense of the primal problem is not correct.

6. Line 130

   The definition of the operator $\mathcal{A}$ seems not consistent with the SDP literature. $\mathcal{A}^*$ is often used to abbreviate the linear matrix inequality.

7. Line 158

   Please be consistent with $z$ and $Z$.

8. Line 240

   The dimension of the matrix is incorrect.

9. Line 250

   The styles of $\text{minimize}$, $\arg \min$ are inconsistent throughout the paper.

10. Line 292

    The definition of $\xi$ is not clear.

11. Line 301

    Matrix Frobenius norm $\\|\cdot\\|_F$ is not defined.

12. Line 420

    The styles of the experiment figures are also inconsistent.

**References**

[1] Boyd, S., Parikh, N., Chu, E., Peleato, B., & Eckstein, J. (2011). Distributed optimization and statistical learning via the alternating direction method of multipliers. *Foundations and Trends® in Machine learning*, *3*(1), 1-122.

[2] Khachiyan, L. G. (1980). Polynomial algorithms in linear programming. *USSR Computational Mathematics and Mathematical Physics*, *20*(1), 53-72.

[3] Tu, S., & Wang, J. (2014). *Practical first order methods for large scale semidefinite programming*. Technical report, Technical report, University of California, Berkeley.

[4] Wen, Z., Goldfarb, D., & Yin, W. (2010). Alternating direction augmented Lagrangian methods for semidefinite programming. *Mathematical Programming Computation*, *2*(3), 203-230.

---

### Official Review · Reviewer_ng4W · 2024-11-03

**Soundness:** 3
**Presentation:** 3
**Contribution:** 2
**Rating:** 6
**Confidence:** 4

**Summary:**

This paper introduces a novel operator stepsize for ADMM, specifically designed to improve scalability in SDP. The proposed stepsize is parameter-free, has a closed form, and is computationally efficient. Empirical results demonstrate substantial improvements in both iteration complexity and runtime, highlighting the practical advantages of this method.

**Strengths:**

* The paper is well-structured and easy to follow.

* They propose the first parameter-free operator stepsize for ADMM, which is both innovative and practical, particularly in the context of SDP.

* The empirical results show substantial speedups compared to traditional scalar stepsize choice, alleviating the SDP scalability issues.

**Weaknesses:**

* Could you provide additional explanation for Equation (3.2)? Deriving the fixed-point view of ADMM from (2.6) is not entirely intuitive.

* What does the variable $\xi$ represent in Lemma 3.1? I think it is better to define it clearly before Lemma 3.1.

* While the algorithm is described as computationally efficient, it would be beneficial to include a formal computational complexity analysis for your method.

* While the experiments effectively demonstrate that the tune-free operator stepsize outperforms traditional scalar stepsizes in ADMM, a comparison with other state-of-the-art algorithms for SDP would provide additional insights into its overall competitiveness.

**Questions:**

See Weakness.

---

### Official Review · Reviewer_CmQo · 2024-11-09

**Soundness:** 2
**Presentation:** 2
**Contribution:** 2
**Rating:** 3
**Confidence:** 4

**Summary:**

This paper presents an operator free alternative direction method of multiplier (ADMM) method for solving semidefinite programming problems. Differ from the standard ADMM approach, the authors claim that the proposed method is tune-free and much more efficient than the naive ADMM method.

**Strengths:**

1. To the best of my knowledge, the proposed method is the first ADMM method that adopts operator stepsize in the context for solving SDP problems.

**Weaknesses:**

1. The notation in the paper is a little bit confusing. For example, $m$ is used to denote the size of the block $\mathbf{1}_1\in\mathbb{S}^m$ in the operator $\mathcal{S}$, but it is also used to denote the number of constraint in the standard dual SDP.
2. While the author claims their method is tune-free, the size of the block $\mathbf{1}_1\in\mathbb{S}^m$ in the operator $\mathcal{S}$ still needs to be tuned. Though the author claims their method works well when setting $m=n-1$ for solving the two test problems in the experimental section, it might not be the case for general SDP problems.
3. I am a little confuse as to why the proposed ADMM method is more scalable than interior point method (IPM). It is well-know that the bottleneck at each iteration of IPM is to solve the $m\times m$ dense Newton subproblem (here, $m$ refer to the number of constraints), which cost $O(m^3)$ time and $O(m^2)$ memory. It seems like the proposed ADMM method has the same per-iteration time and memory complexity because the update equation for $x$ in Algorithm 1 (and $X$ in algorithm 2) required inverting an $m\times m$ matrix $\tilde{A}\tilde{A}^T$ (and $(m+n)\times (m+n)$ KKT system) at each iteration, which also cost $O(m^3)$ time and $O(m^2)$ memory. Unlike the update equation for $X$ in the original ADMM method (1.5), in which the $m\times m$ matrix $\bar{A}\bar{A}^T$ is fixed and its inverse (or Cholesky factor) only needs to be computed once.
4. Based on point 3, it seems like the per-iteration cost of IPM and the proposed ADMM method are same. However, IPM only requires $O(\sqrt{n}\log(1/\epsilon))$ iteration to reach $\epsilon$ accuracy, which is in general significantly less than the amount iteration required for ADMM to reach the same accuracy.

**Questions:**

1. It seems like the test problems considered in the experimental section are too simple; these are the SDP problems that only have $m=O(n)$ constraints, and have the structure where $\mathbf{AA}^T$ is diagonal and easy to invert. Would the proposed ADMM still be efficient for the SDP problems that have  $m=O(n^2)$ constraints and dense  $\mathbf{AA}^T$? For instance, Lyapunov stability problem $\min_{X\succeq 0}\ 0 : XP_i+P_i^TX\preceq 0\ \forall i$.
2. If we set the size of the block $\mathbf{1}_1\in\mathbb{S}^m$ in the operator $\mathcal{S}$ to be $n$, would the proposed method reduce back to the original ADMM method described in section 1.3.1?

---

### Meta-Review · Area_Chair_dFjj · 2024-12-19

**Metareview:**

This paper introduces a novel operator stepsize for ADMM, specifically designed to improve scalability in SDP. The proposed stepsize is parameter-free, has a closed form, and is computationally efficient. Empirical results demonstrate substantial improvements in both iteration complexity and runtime, highlighting the practical advantages of this method.

Unfortunately, two reviewers critically point out that the paper does not truly overcome the bottleneck in existing methods, as it still relies on projection and linear system solving. One reviewer further points out that the per-iteration cost of IPM and the proposed ADMM are the same, but IPM converges in far fewer iterations.

In my opinion, this hole significantly undermines the claimed contribution of the work. The authors did not post a rebuttal, so the concern remains unaddressed.

**Additional Comments On Reviewer Discussion:**

The authors did not post a rebuttal, so no further discussion was necessary.

---

### Decision · Program_Chairs · 2025-01-22

Reject